# NEURAL CAUSAL GRAPH FOR INTERPRETABLE AND INTERVENABLE CLASSIFICATION

**Jiawei Wang[1], Shaofei Lu[1],[*] Da Cao[1], Dongyu Wang[2],**
**Yuquan Le[1], Zhe Quan[1], Tat-Seng Chua[3]**
[1]Hunan University, Changsha, China
[2]University of Science and Technology of China, Hefei, China
[3]National University of Singapore, Singapore
{wangjiawei0531,caoda0721,dywang3254}@gmail.com
{sflu,leyuquan,quanzhe}@hnu.edu.cn,dcscts@nus.edu.sg

## ABSTRACT

Advancements in neural networks have significantly enhanced the performance of classification models, achieving remarkable accuracy across diverse datasets. However, these models often lack transparency and do not support interactive reasoning with human users, which are essential attributes for applications that require trust and user engagement. To overcome these limitations, we introduce an innovative framework, Neural Causal Graph (NCG), that integrates causal inference with neural networks to enable interpretable and intervenable reasoning. We then propose an intervention training method to model the intervention probability of the prediction, serving as a contextual prompt to facilitate the fine-grained reasoning and human-AI interaction abilities of NCG. Our experiments show that the proposed framework significantly enhances the performance of traditional classification baselines. Furthermore, NCG achieves nearly 95% top-1 accuracy on the ImageNet dataset by employing a test-time intervention method. This framework not only supports sophisticated post-hoc interpretation but also enables dynamic human-AI interactions, significantly improving the model's transparency and applicability in real-world scenarios.

## 1 INTRODUCTION

In the realm of deep neural networks, classification tasks frequently serve as benchmarks for evaluating the performance and robustness of emerging models (Krizhevsky et al., 2012; Wang et al., 2022; Liu et al., 2024; Lv et al., 2025b). These models, particularly those based on foundation models (Vaswani et al., 2017; Bommasani et al., 2021; Lv et al., 2025c), are powerful in handling complex datasets (Deng et al., 2009; Wang et al., 2019; Schuhmann et al., 2022; Wang et al., 2023; Shen et al., 2024a;b; Wang et al., 2024a; Lv et al., 2025a) but often lack transparency and interactivity, which are key attributes for applications requiring high levels of trust and user engagement. Traditional approaches have focused on post-hoc interpretability techniques, such as discriminative feature visualization (Zhou et al., 2016; Selvaraju et al., 2017; Zhu et al., 2021), to elucidate the decision-making processes of models. However, these models primarily function as black boxes, lacking the ability for interactive interventions. This limitation hinders alignment with human reasoning and prevents real-time adaptation of the model behavior. Furthermore, causality-driven graph neural networks (Jiang et al., 2023; Job et al., 2023) focus on interpreting data by quantifying cause-effect relationships among variables or recovering the causal graph from observed datasets. However, these models lack the capability for interactive intervention and do not support interactive classification involving human users.

To address these challenges, we propose the Neural Causal Graph (NCG), a novel framework that enable intervenable classification by integrating causal inference within neural networks architectures. Unlike conventional methods that only provide post-hoc interpretability, NCG constructs a causal graph modeling the data's underlying mechanisms, enabling the model to reason through causal

---

*Corresponding Author

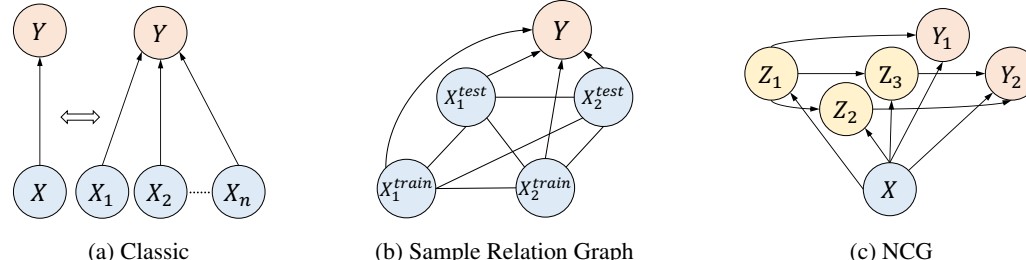

Figure 1: The graphical overview of three distinct classification paradigms. Figure (a) illustrates a traditional classification approach that treats the label prediction process of each sample as independent. Figure (b) depicts a transductive learning method based on the relation (undirected) graph of different samples. In contrast, Figure (c) presents the neural causal graph approach, where the model constructs a graph with prior concept nodes $Z_i$ and posterior concept nodes $Y_i$, with edges representing causal relationships among various concepts.

relationships and support intervenable classification. This NCG paradigm differs from both the traditional classification approaches (Simonyan & Zisserman, 2014; He et al., 2016; Zhou et al., 2023) and methods relying on sample relation graphs (Liu et al., 2018; Ciano et al., 2021; Wang et al., 2024b), as illustrated in Figure 1. Central to our approach is the utilization of a structural causal model (SCM) (Pearl, 2000; Peters et al., 2017) that serves as the backbone of the NCG. This causal framework allows for the explicit modeling and manipulation of causal relationships among a set of concept nodes, which represent abstracted concepts or classes in the dataset. By integrating SCM with deep neural networks, NCG is capable of performing intervenable reasoning, where users can actively manipulate certain concepts (do-interventions) to observe how these changes influence the outcomes, providing a powerful tool for "what-if" analysis (Pearl, 2000).

Specifically, our method employs an efficient algorithm for estimating the causal effects within the NCG by leveraging do-calculus (Pearl, 2012) and the potential outcome model (POM) (Rosenbaum & Rubin, 1983; Pearl, 2012). This allows NCG to infer causal effects accurately without the biases typically introduced by confounding variables from the input data. Moreover, the framework is equipped with an intervention training method that not only enhances model training with augmented causal reasoning capabilities but also facilitates dynamic interactions during model inference. These interactions enable users to guide the model's reasoning process, making the AI system more transparent and adaptable in practical scenarios.

We validate the effectiveness of the NCG through extensive experiments on both the ImageNet dataset and a custom causal-related dataset. Our results demonstrate that NCG surpasses existing classification models in terms of both accuracy and robustness. Additionally, we showcase the test-time intervention mechanism of the NCG model, which achieves nearly 95% top-1 accuracy on the ImageNet dataset. This capability not only supports post-hoc interpretation but also facilitates intervention processes, allowing for enhanced interactions informed by further human knowledge.

The main contributions of this work are summarized as follows:

- We introduce a novel classification paradigm that combines causal inference with neural networks to enhance the interpretability and interactivity of classification models.

- We develop a novel algorithm for unbiased causal effect estimation within NCG construction and propose an intervention training method, which allows for test-time intervention on the models.

- Extensive experiments are conducted on the ImageNet dataset and a self-collected causal-related dataset Bird, which demonstrate the rationality and effectiveness of the NCG method. Additionally, we make the datasets and implementations available to the research community to facilitate further research[1].

---

[1] https://github.com/JaveyWang/NCG

## 2 RELATED WORKS

**Causal Inference.** Causal inference frameworks, such as SCM and POM, empower researchers to analyze existing observational data without necessitating extensive interventional studies like randomized control trials (Rosenbaum & Rubin, 1983; Pearl, 2000; Bareinboim & Pearl, 2014; Bareinboim et al., 2015). The SCM defines a directed graph representing causal relationships among system variables. This allows for the virtual estimation of interventional experiments using expressions like $P(Y|do(X))$, where $do(\cdot)$ denotes a controlled experiment on the real-world variable $X$. The do-calculus rules have been introduced to transform interventional queries into association expressions that do not include the $do$ operator, achieved by manipulating the causal graph with consideration of conditional independence property (Pearl, 2000). On the other hand, potential outcome model, also known as the Rubin causal model (Rosenbaum & Rubin, 1983), offers a general causal framework for estimating population-level causal effects, often called average treatment effects. In our NCG method, we provide an unbiased estimation formula to calculate the causal effects of NCG by integrating both the do-calculus rules from SCM and the propensity score technique from POM.

**Classification Model.** Recent advances in classification models have been driven by deep neural networks and the application of the softmax operator, which have greatly enhanced the ability to learn discriminative features (Krizhevsky et al., 2012; Szegedy et al., 2013; Zeiler & Fergus, 2014; Zhou et al., 2023; Ma et al., 2023). Recent methodologies have implemented causal interventions on task-level causal graphs, aiming to develop networks that learn without the bias of spurious correlations (Wang et al., 2021; Yue et al., 2021; Rao et al., 2021; Qin et al., 2021; Li et al., 2023; Ding et al., 2024). However, these methods exhibit limitations as they retain the label independence when performing representation learning, thereby hindering the ability to understand relationships among labels. In contrast, our NCG framework directly exploits causal information through message propagation on a directed label graph, thus supporting interventional reasoning at the conceptual level instead of the task level.

## 3 METHOD

We present the NCG framework, as depicted in Figure 2, which comprises two stages: 1) constructing and estimating the neural causal graph; and 2) performing concept construction and concept reasoning. In the following sections, we first present the preliminaries to establish the relationship between SCM and NCG (Section 3.1). We then detail the construction and estimation processes of the NCG (Section 3.2), followed by an detailed implementation of the reasoning and training processes built upon it (Section 3.3).

### 3.1 PRELIMINARIES

To effectively utilize the causal information in neural causal graph, we build upon the SCM framework (Pearl, 2000). Our SCM is defined as a 4-tuple $\mathcal{M} = \langle \mathbf{U}, \mathbf{C}, \mathcal{F}, P(\mathbf{U}) \rangle$. Here, $\mathbf{U}$ is a set of exogenous variables that represent unobservable factors external to the model, and $\mathbf{C} = \{\mathbf{C}_1, \mathbf{C}_2, \ldots, \mathbf{C}_n\}$ is a set of endogenous variables (concepts) that are determined by the relationships among variables in $\mathbf{U} \cup \mathbf{C}$. Each endogenous variable $\mathbf{C}_j$ depends on its parent variables $\text{Pa}(\mathbf{C}_j) \subseteq \mathbf{C}$ and its associated exogenous variable $\mathbf{U}_j$. The structural functions $\mathcal{F} = \{f_{\mathbf{C}_1}, f_{\mathbf{C}_2}, \ldots, f_{\mathbf{C}_n}\}$ are deterministic mappings, where each $f_{\mathbf{C}_j}$ determines $\mathbf{C}_j$ based on $\text{Pa}(\mathbf{C}_j)$ and $\mathbf{U}_j$, i.e., $\mathbf{C}_j \leftarrow f_{\mathbf{C}_j}(\text{Pa}(\mathbf{C}_j), \mathbf{U}_j)$. Finally, $P(\mathbf{U})$ defines the joint probability distribution over the exogenous variables $\mathbf{U}$.

In the context of NCGs, the SCM framework provides a principled way to model the relationships between concepts and their interactions. Specifically, the endogenous variables $\mathbf{C}$ represent concept nodes in the NCG, the exogenous variables $\mathbf{U}$ capture external factors affecting the concepts, and the structural functions $\mathcal{F}$ govern how causal information propagates across the concept graph. By leveraging this framework, NCGs enable robust graph reasoning, supporting both interpretable and intervenable classification.

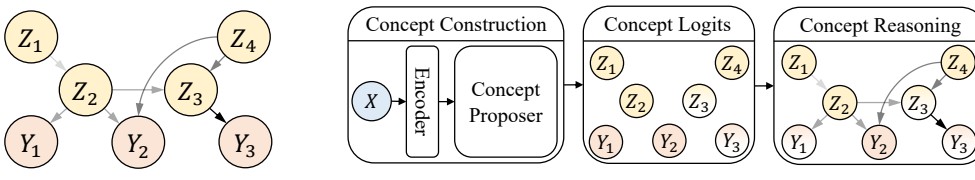

(a) Neural Causal Graph          (b) Concept Construction and Reasoning

Figure 2: The illustration of our proposed NCG framework. Figure (a) depicts the construction and estimation of the NCG with prior concepts denoted as $Z_i$ and posterior concepts as $Y_j$. Different shades of gray on the edges represent varying causal effects among the concepts. Figure (b) shows how the model classifies an input $X$. For simplification, all edges where the sample variable $X$ points to each node are omitted.

## 3.2 NEURAL CAUSAL GRAPH CONSTRUCTION AND ESTIMATION

### 3.2.1 GRAPH CONSTRUCTION

We assume most real-world problems have underlying knowledge structures which can be used to form a causal graph. To construct NCGs, we initially preprocess the classification dataset using WordNet (Miller, 1995) to extract concepts and causal relationships, in which WordNet is a directed acyclic graph that includes a vast number of words and lexical relations. Given a classification dataset $\mathcal{D} = \{(X_1, y_1), (X_2, y_2), ..., (X_m, y_m)\}$, where $X_i \in \mathcal{X}$ represents an input, and $y_i \in \mathcal{Y} = \{Y_1, Y_2, ..., Y_n\}$ denotes the corresponding label from the label set $\mathcal{Y}$ given examples of $(X_i, y_i)$. We extract all labels $Y_i$ from WordNet and consider them as the posterior concepts of NCG. We then identify the ancestor nodes $\mathcal{Z} = \{Z_1, Z_2, ..., Z_c\}$ of the labels in WordNet as prior concepts and retain the corresponding directed edges in WordNet as the node relationships of NCG. To control the size of NCG, we enforce each prior concept to be connected to at least one posterior concept within a maximum of five hops. Finally, the NCG for a given dataset is formulated as a subgraph $\mathcal{G} = \{\mathcal{N}, \mathcal{E}\}$, which is extracted from WordNet. Here, $\mathcal{N} = \mathcal{Z} \cup \mathcal{Y} \in \{0, 1\}^{c+n}$ encompasses both the prior concepts $\mathcal{Z}$ and the posterior concepts (or labels) $\mathcal{Y}$, while $\mathcal{E}$ represents the causal edges connecting these concepts. To facilitate subsequent reasoning processes, all elements of $\mathcal{N}$ are arranged in topological order, details of which will be discussed in Figure 6.

### 3.2.2 MULTI-LABEL ASSIGNMENT

In traditional multi-class classification, a typical visual task assumes that each sample has only one ground-truth label. However, in our NCG paradigm, we transform the task into a multi-label classification task to better capture concept information. For a given example $(X, y)$ in the dataset, we define all ancestor nodes of the label $y$ in the NCG as $A = \{a | y \leq a\}$, where $\leq$ signifies the reachability relation, indicating the existence of paths from $a$ to $y$ in the directed acyclic graph. Consequently, the label $y$ and its ancestor nodes $A$ are considered as the conceptual labels $\tilde{Y} = (A \cup y) \subset \mathcal{N}$ for the example $X$. The appendix presents a detailed overview of the labels and their ancestor nodes, as illustrated in Figure 5.

### 3.2.3 CAUSAL INTERVENTION AND ESTIMATION

Assuming that the neural causal graph can be defined as a linear structural equation model (Pearl, 2000), estimating the causal effect between nodes becomes a challenging task due to the presence of a common confounding factor $X$. For example, as illustrated in Figure 1(c), each concept not only relies on their neighborhood concepts in $\mathcal{G}$, but also primarily depends on a particular sample $X$. This implies that attempting to estimate the conditional distribution $P(C'|C)$ of an edge $C \rightarrow C'$ would only describe the correlation rather than the causality, as shown in Figure 3(a). The sample variable $X$ acts as a confounder for both nodes $(C, C')$, leading to an open path $C \leftarrow X \rightarrow C'$, which makes it difficult to ascertain the actual causal effect of $C'$ caused by $C$. Therefore, the key to estimating the direct causal effect between these nodes lies in mitigating the impact of spurious correlations between $C$ and $C'$. We aim to address this problem by leveraging the backdoor criteria of the SCM (Pearl, 2000). We replace $P(C'|C)$ with $P(C'|do(C))$ and block the influence of path $C \leftarrow X \rightarrow C'$ by

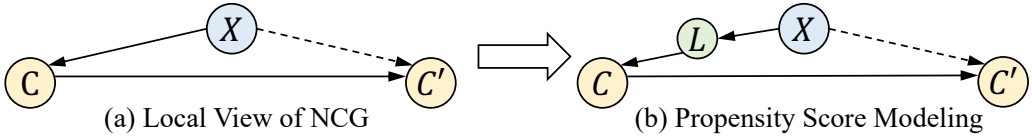

Figure 3: Illustration of using propensity scores to estimate causal effects in the NCG. Solid arrows indicate direct edges, while dashed arrows signify the presence of at least one unblocked path between $X$ and $C'$. Figure (a) shows a local view of NCG where the sample variable $X$ acts as a confounder of nodes $C$ and $C'$. Figure (b) introduces a propensity score variable, allowing $C$ to be directly dependent on a real scalar $L := L(X)$, rather than on the sample variable $X$.

adjusting the variable $X$, which is formulated as:

$$P(C'|do(C)) = \sum_X P(C'|C, X)P(X),\tag{1}$$

where $do(\cdot)$ represents an intervention operation that follows do-calculus (Pearl, 2000) technique to achieve an unbiased estimation. It is worth noting that $C$ and $C'$ are both binary variables, while $X$ is a high dimension variable, which poses difficulties in estimating Equation (1). This challenge is also known as the curse of dimensionality problem (Friedman, 1997; Indyk & Motwani, 1998; Zhang & Zhou, 2010), commonly encountered when applying statistical theory to unstructured data. Therefore, reducing the dimension of $X$ becomes crucial to achieve a more accurate estimation of Equation (1). Fortunately, this problem can be effectively addressed by introducing a propensity score $L$ between $X$ and $C$, as shown in Figure 3(b). This induces a conditional distribution $P(C'|C, L)$ that substantially has lower dimension:

$$P(C'|do(C)) = \sum_L P(C'|C, L)P(L),\tag{2}$$

where the equation holds because of the backdoor criterion (Pearl, 2000; 2012) applied in Figure 3(b). This NCG estimation scenario establishes a connection between the SCM framework and the POM framework, and Equation (2) points out an effective estimation form of the causal graph.

### 3.2.4 CAUSAL EFFECT CALCULATION

To estimate the causal effects of edges in the neural causal graph, we employ two classical methods from the POM framework: Propensity Score Matching (PSM) (Rosenbaum & Rubin, 1983; Stuart, 2010) and Doubly Robust Learning (DRL) (Bang & Robins, 2005; Kang & Schafer, 2007; Cao et al., 2009). Both methods compute the average treatment effect (ATE) for each edge, which quantifies the causal influence of a source node (treatment) on a target node (outcome) in the NCG.

**Propensity Score Matching** balances treatment and control groups by matching samples with similar propensity scores (Rosenbaum & Rubin, 1983; Stuart, 2010), which represent the probability of treatment given background variables. This enables counterfactual inference by comparing outcomes between matched groups.

**Doubly Robust Learning** enhances the estimation by combining inverse probability weighting and regression-based methods. This approach ensures robustness against misspecification in either the propensity score or the outcome model, enabling effective estimation of counterfactual outcomes for all samples.

The causal effects computed using PSM and DRL are stored in the NCG adjacency matrix $A \in \mathbb{R}^{n \times n}$. Each entry $A_{ij}$ represents the average treatment effect from the source node $i$ (treatment) to the target node $j$ (outcome). If there is no edge between nodes $i$ and $j$, $A_{ij}$ is set to zero. This adjacency matrix serves as the foundation for reasoning and training in subsequent stages of the NCG framework. For more details on the implementation of PSM and DRL, please refer to Appendix A.3.

### 3.3 MODEL IMPLEMENTATION

### 3.3.1 CONCEPT CONSTRUCTION

In Figure 2(b), the encoder represents a pre-trained network that generates a feature vector $\mathbf{h} = h(X)$. The concept proposer, denoted as a neural network $g(\cdot)$, takes the feature $\mathbf{h} \in \mathbb{R}^d$ as an input and

produces concept logits $\mathbf{O} \in \mathbb{R}^{n \times d}$ for concept reasoning. Here, $n$ denotes the number of concept nodes in NCG, and $d$ represents the number of heads of NCG. When $d = 1$, the concept construction process is consistent with the traditional classification paradigm, as each logit directly corresponds to one concept.

### 3.3.2 CONCEPT REASONING

According to the definition of SCM as illustrated in the Section 3.1, for each concept variable $\mathbf{C}_j \in \mathbf{C}$, its reasoning function can be represented as $\mathbf{c}_j = f_{\mathbf{C}_j}(X, Pa(\mathbf{C}_j), \mathbf{U}_{\mathbf{C}_j})$, where $\mathbf{c}_j \in \mathbb{R}^d$ is a vector determined by its parents $Pa(\mathbf{C}_j)$ and $\mathbf{U}_{\mathbf{C}_j}$ corresponds to an exogenous variable related to $\mathbf{C}_j$. The function $f_{\mathbf{C}_j}$ is a structural equation that aggregates and updates messages from neighborhoods. Formally, the formulation is shown as follows:

$$\mathbf{c}_j = \phi(g(h(X))_j + \sum_i A_{ij} \mathbf{c}_i + \mathbf{U}_{\mathbf{C}_j}), \tag{3}$$

where $\mathbf{C}_i \in Pa(\mathbf{C}_j)$ and the subscripts $i, j$ are used as index operations that access the corresponding concept nodes, and $A_{ij}$ represents the directed causal weight from $\mathbf{C}_i$ to $\mathbf{C}_j$, as discussed in Section 3.2.4. $\mathbf{U}_{\mathbf{C}_j}$ is defined as an exogenous variable $\mathbf{U}_{\mathbf{C}_j} \sim \mathcal{N}(0, 1)$. $\phi$ is a update function, typically set as an identical mapping to form a linear model. Finally, by applying the reasoning function in the topological order of the NCG for each node, we can transform the initial concept logits $\mathbf{O} \in \mathbb{R}^{n \times d}$ into updated concept logits $\mathbf{O}' \in \mathbb{R}^{n \times d}$.

### 3.3.3 LOSS CALCULATION

After the process of concept reasoning, we calculate the loss for optimization by using the binary cross entropy as follows:

$$\begin{cases} \mathcal{L} = -\frac{1}{n} \sum_{i=1}^{n} \left[ \mathbf{Y}_i \log \hat{\mathbf{Y}}_i + (1 - \mathbf{Y}_i) \log(1 - \hat{\mathbf{Y}}_i) \right] \\ \hat{\mathbf{Y}}_i = \sigma(\mathbf{O}'_i \mathbf{w}_i), \end{cases} \tag{4}$$

where $\mathbf{Y}_i \in \{0, 1\}$ represents the ground-truth label of the $i^{th}$ node in NCG, and $n$ denotes the total number of concept nodes. $\mathbf{w}_i \in \mathbb{R}^{d \times 1}$ is a learnable scaled weight of the $i^{th}$ node that is initialized as $1/d$ for each head. It transforms the updated concept logits to a score, and $\sigma$ is the sigmoid function that produces a predicted probability for each concept node.

### 3.3.4 INTERVENTION TRAINING METHOD

To enable the model to perform test-time interventions by answering questions like "What would Y be if we do Z", commonly referred to as "What if" question (Pearl, 2000), we propose the intervention training method that equips the model to reason based on intervened concepts. This approach is designed to enhance the test-time intervention ability by aligning the training and testing distributions, ensuring robustness under test-time interventions. Inspired by the principles of structural causal models, this approach simulates the effects of the $do(\cdot)$ operator during training, isolating the influence of intervened nodes by eliminating their dependence on parent nodes. Specifically, during each training iteration, a subset of prior concepts is randomly intervened with an intervention rate $p = 0.15$, and their concept logits are fixed using the formula $(2\mathbf{Y}_i - 1) \times v$. $\mathbf{Y}_i \in \{0, 1\}$ represents the corresponding multi-label value, and $v$ is an empirically determined confidence value set to 5. The sigmoid output of $-5$ or $5$ reflects high certainty about the node's intervention status.

This method injects reasoning capabilities into the model by explicitly forcing it to learn the dynamics among concepts while addressing "What if" questions. By simulating test-time scenarios, the model can be conditioned to predict posterior concepts based on both inferred and intervened prior concepts. An overview of the procedure for intervention training is detailed in Algorithm 2.

## 4 EXPERIMENTS

### 4.1 EXPERIMENTAL SETTINGS

**Datasets.** Two datasets are used in this study: Bird and ImageNet (Deng et al., 2009). The Bird dataset is a self-collected and more interpretable subset of the larger ImageNet dataset. Table 1

Table 1: The statistics of two datasets, including the number of prior and posterior concepts, and the size of training set and testing set.

| Dataset | Prior | Posterior | Training set | Testing Set |
|---|---|---|---|---|
| Bird | 16 | 9 | 11,700 | 450 |
| ImageNet | 1,357 | 1,000 | 1,281,167 | 50,000 |

Table 2: The overall performance comparison of various methods on the Bird and ImageNet datasets. Each row represents a specific model performed. The best performance for Acc and F1 is highlighted in bold. The baseline model for t-test is Multi-class. (Section 4.2)

| Dataset | Backbone | Model | Acc | Std | p-value | F1 | Std | p-value |
|---|---|---|---|---|---|---|---|---|
| Bird | ResNet50 | Multi-class | 90.89 | 0.92 | - | 90.81 | 0.95 | - |
| | | Multi-label | 91.33 | 1.28 | 0.5897 | 91.32 | 1.26 | 0.5383 |
| | | CBM | 90.22 | 1.10 | 0.3814 | 90.06 | 1.21 | 0.3585 |
| | | NCG (PSM) | 92.31 | 0.36 | 0.0207 | 92.29 | 0.37 | 0.0205 |
| | | NCG (DRL) | **93.42** | 0.11 | 0.0006 | **93.41** | 0.11 | 0.0006 |
| | CLIP | Multi-class | 93.33 | 0.63 | - | 93.31 | 0.67 | - |
| | | Multi-label | 93.16 | 0.74 | 0.7349 | 93.15 | 0.82 | 0.7713 |
| | | CBM | 90.67 | 1.10 | 0.0030 | 90.64 | 1.22 | 0.0049 |
| | | NCG (PSM) | 94.22 | 0.20 | 0.0270 | 94.22 | 0.20 | 0.0315 |
| | | NCG (DRL) | **94.49** | 0.17 | 0.0074 | **94.53** | 0.17 | 0.0081 |
| ImageNet | ResNet50 | Multi-class | 73.08 | 0.09 | - | 72.35 | 0.11 | - |
| | | Multi-label | 73.23 | 0.05 | 0.1882 | 72.52 | 0.06 | 0.2256 |
| | | CBM | 73.22 | 0.55 | 0.7545 | 72.78 | 0.70 | 0.4736 |
| | | NCG (PSM) | **73.75** | 0.02 | 0.0035 | **73.35** | 0.11 | 0.0046 |
| | | NCG (DRL) | 73.73 | 0.04 | 0.0045 | 73.24 | 0.03 | 0.0033 |
| | CLIP | Multi-class | 83.49 | 0.06 | - | 82.97 | 0.06 | - |
| | | Multi-label | 83.58 | 0.20 | 0.5744 | 83.05 | 0.16 | 0.5466 |
| | | CBM | 83.32 | 0.79 | 0.7847 | 82.90 | 0.98 | 0.9290 |
| | | NCG (PSM) | **84.44** | 0.12 | 0.0029 | **83.98** | 0.14 | 0.0032 |
| | | NCG (DRL) | 83.98 | 0.08 | 0.0098 | 83.35 | 0.14 | 0.0508 |

provides the statistics for both datasets. Given the limited size of the Bird dataset, multiple repetitions of experiments can be conducted to ensure the reliability of the model performance. NCGs for both two datasets are constructed using the previously described process, and a graphical overview for the Bird dataset is provided in Figure 5.

**Evaluation Metrics.** The performance of the model is evaluated using two widely applied multi-classification metrics (Pedregosa et al., 2011): accuracy (Acc) and macro-F1 score (F1). Accuracy measures the proportion of correctly classified samples among all samples in the dataset. It provides a general overview of the model's overall performance. Macro-F1 score calculates the harmonic mean of precision and recall for each class, and then takes the average over all classes. It gives a balanced assessment of the model performance across all classes. Note that for the Bird dataset and ImageNet dataset, we repeat each experiment 5 times and 3 times, respectively, and the average results are reported. We calculate standard deviation (Std) to indicate the variation in results of Acc and F1. P-values are also provided using a two-sample t-test, with p-values less than 0.05 indicating a statistically significant difference compared to the baseline model in different experiments.

**Baselines.** Our NCG framework can be viewed as a plug-in module built upon traditional classification methods, so we choose multi-class and multi-label classification settings as baselines. The multi-class baselines include training the model using only posterior concepts, while the multi-label baseline involve using both the prior and posterior concepts for model optimization. Additionally, we utilize a Concept Bottleneck Model (CBM) (Koh et al., 2020) as another baseline. The CBM predicts prior concepts through a bottleneck layer, and then uses these concepts to predict the label. This design allows for intervention by editing prior concept values and propagating these changes to the final posterior concept values.

**Implementation Details.** To demonstrate the effectiveness of our proposed framework, we build our NCG model using two frozen pre-trained backbones, ResNet50 (He et al., 2016) and ViT from

Table 3: Experimental results of the NCG framework using different causal weights on the Bird dataset. The best performance for Acc and F1 is highlighted in bold. The baseline model for t-test is One. (Section 4.3)

| Backbone | Weight | Acc | Std | p-value | F1 | Std | p-value |
|---|---|---|---|---|---|---|---|
| ResNet50 | One | 91.07 | 0.50 | - | 91.21 | 0.40 | - |
| | Random | 90.85 | 1.80 | 0.8179 | 90.81 | 1.85 | 0.6811 |
| | Zero | 91.96 | 0.95 | 0.1357 | 91.92 | 1.00 | 0.2237 |
| | Learn | 91.87 | 0.23 | 0.0192 | 91.80 | 0.23 | 0.0329 |
| | PSM | 92.31 | 0.36 | 0.0037 | 92.29 | 0.37 | 0.0044 |
| | DRL | **93.42** | 0.11 | $< 0.0001$ | **93.41** | 0.11 | $< 0.0001$ |
| CLIP | One | 92.76 | 0.23 | - | 93.02 | 0.20 | - |
| | Random | 92.51 | 1.00 | 0.6482 | 92.49 | 0.99 | 0.3226 |
| | Zero | 93.69 | 0.30 | 0.0011 | 93.69 | 0.29 | 0.0053 |
| | Learn | 94.18 | 0.26 | $< 0.0001$ | 94.19 | 0.25 | $< 0.0001$ |
| | PSM | 94.22 | 0.20 | 0.0270 | 94.22 | 0.20 | $< 0.0001$ |
| | DRL | **94.49** | 0.17 | $< 0.0001$ | **94.53** | 0.17 | $< 0.0001$ |

OpenCLIP-H-14[2] (Radford et al., 2021). Recognizing the disparity in complexity between the two datasets, we employ a linear layer perceptron as the concept proposer $g$ for the Bird dataset and and a two-layer perceptron (with intermediate channels equal to input channels) for the ImageNet dataset. For the ImageNet dataset, we also utilize an eight-head NCG ($d = 8$) and set the update function $\phi$ as a three-layer perceptron with $64$ intermediate channels, incorporating $tanh$ activation functions in the first two layers to regulate the scope of the final output.

Our framework is implemented based on PyTorch[3] and DGL[4], and all models are trained with a mini-batch size of 512 on a machine equipped with four Nvidia-3090Ti GPUs. We use AdamW (Kingma & Ba, 2014; Loshchilov & Hutter, 2017) as the optimizer with a learning rate of 0.01 for the Bird dataset and 0.001 for the ImageNet dataset. We train the models for 50 epochs for the Bird dataset and 3 epochs for the ImageNet dataset without any data augmentation.

## 4.2 OVERALL PERFORMANCE COMPARISON

To evaluate the effectiveness of our proposed method, we conduct several experiments using two different backbones on the two datasets to compare the performance of our proposed framework against baseline approaches.

Table 2 demonstrates that the two proposed NCGs outperform other baselines on different datasets and backbone models. We have the following observations: 1) NCGs outperform Multi-class and Multi-label methods. We argue that this is because NCG models learn the causal relationships among concepts, which gives them better generalization ability; 2) The performance of NCGs is significantly better than that of CBM. We attribute this to the correct concept structure introduced by the causal graph, whereas CBM only separates the prior concepts and posterior concepts without explicitly modeling their causal relationships; 3) Interestingly, it is important to note that NCG models can even improve the performance of ResNet50 (NCG vs. Multi-class), a model that is already pre-trained on ImageNet. This indicates that valuable information for classification is derived from the NCG's causal structure and weights.

## 4.3 EFFECT OF CAUSAL WEIGHTS

To investigate the importance of causal weights, we devise six variants of the NCG model with different causal weight implementations, including PSM, DRL, Learn, One, Random and Zero. PSM and DRL are unbiased estimation methods based on the potential outcome model. One and Zero methods refer to the weights being set as a constant one or zero for each edge, respectively. Random denotes a randomized weight value from zero to one, while Learn implies that the weight is initialized as one and is learnable during training.

---

[2]https://laion.ai/blog/giant-openclip/
[3]https://pytorch.org
[4]https://www.dgl.ai

Table 4: The experimental results of using different components based on ResNet50 and CLIP backbones. Note that IT and LSW are the abbreviation of Intervention Training and Learnable Scaled Weight, respectively. The best performance for Acc and F1 is highlighted in bold. The baseline model for t-test is w/o IT and LSW. (Section 4.4)

| Backbone | Model | Components | | Acc | | | F1 | | |
|---|---|---|---|---|---|---|---|---|---|
| | | IT | LSW | Value | Std | p-value | Value | Std | p-value |
| ResNet50 | NCG (PSM) | ✓ | ✓ | **92.31** | 0.36 | 0.0216 | **92.29** | 0.37 | 0.0296 |
| | | ✓ | | 91.29 | 0.62 | 0.0952 | 91.18 | 0.60 | 0.1112 |
| | | | ✓ | 90.89 | 1.53 | 0.2237 | 90.86 | 1.57 | 0.2083 |
| | | | | 89.11 | 0.36 | - | 88.75 | 0.37 | - |
| | NCG (DRL) | ✓ | ✓ | **93.42** | 0.11 | 0.0042 | **93.41** | 0.11 | 0.0067 |
| | | ✓ | | 92.09 | 0.30 | 0.0470 | 92.05 | 0.29 | 0.0631 |
| | | | ✓ | 91.38 | 0.88 | 0.2111 | 91.41 | 0.89 | 0.2093 |
| | | | | 90.09 | 1.68 | - | 90.00 | 1.88 | - |
| CLIP | NCG (PSM) | ✓ | ✓ | **94.22** | 0.20 | 0.0085 | **94.22** | 0.20 | 0.0079 |
| | | ✓ | | 93.29 | 0.41 | 0.3728 | 93.27 | 0.42 | 0.3859 |
| | | | ✓ | 93.47 | 0.33 | 0.1924 | 93.46 | 0.33 | 0.1901 |
| | | | | 92.89 | 0.20 | - | 92.88 | 0.20 | - |
| | NCG (DRL) | ✓ | ✓ | **94.49** | 0.17 | 0.0088 | **94.53** | 0.17 | 0.0073 |
| | | ✓ | | 94.44 | 0.14 | 0.0100 | 94.41 | 0.15 | 0.0106 |
| | | | ✓ | 93.69 | 0.27 | 0.1535 | 93.70 | 0.26 | 0.1400 |
| | | | | 92.98 | 0.17 | - | 92.96 | 0.17 | - |

As shown in Table 3, we have three observations: 1) The performance of PSM and DRL methods is superior to that of other methods on both backbone models, indicating that the estimated causal weight captures the relationships among concepts. 2) Methods with constant weights, such as One, Random and Zero, suffer from incorrect weight assignments and exhibit higher errors on both backbones. This emphasizes that accurate causal weight estimation is crucial for optimization and reasoning of NCG. 3) The Learn method demonstrates moderate success with CLIP but is inadequate with ResNet50, exposing training instability and the inherent difficulty in directly learning causal weight between concepts. This further validates the effectiveness of our proposed PSM and DRL estimation methods.

## 4.4 ABLATION STUDIES

We conduct ablation studies on our NCG framework to explore the impact of intervention training and learnable scaled weight, which are specially designed for optimization and reasoning within our NCG framework. As shown in Table 4, we have the following observations: 1) Utilizing both intervention training and learnable scaled weight significantly enhances the NCG framework's reasoning ability across two backbones and two distinct causal weights. Without these elements, the performance decreases considerably, validating the efficacy of these design features. 2) The intervention training component alone greatly improves the reasoning ability of the NCG model, as evidenced by a marked improvement of nearly 2% over the worst model. Such improvement can be attributed to causal intervention, which compels the model to address "What If" queries within the NCG, thereby fostering reasoning skills among various concepts. Furthermore, the NCG can be treated as a world model about a dataset, encompassing causal knowledge about different concepts. Therefore, intervening on the world model creates many out-of-distribution examples, which can be viewed as a data augmentation method and thus facilitating the learning of NCG's generalization ability. 3) Compared to models lacking both components, the incorporation of learnable scaled weight alone yields a significant improvement. It is because those prior concepts that near posterior concepts may have much larger updated logits because their values are accumulated from their ancestors, which results in the label imbalance problem. Therefore, the learnable scaled weight can reweight different updated logits and achieve a more balanced loss.

## 4.5 TEST-TIME INTERVENTION

We propose a test-time intervention experiment to harness the powerful potential of human-AI collaboration in our model. By enabling intervention on prior concepts, human users can interactively

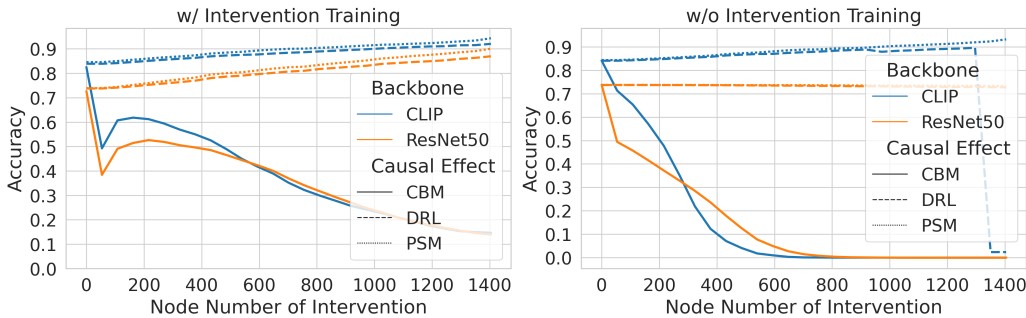

Figure 4: The test-time intervention accuracy of various models on the ImageNet dataset. (Section 4.5)

guide the model's behavior during testing. Through a series of controlled experiments, we explore how incremental interventions across different segments of prior concepts influence the accuracy of posterior concepts. Specifically, we investigate the intervention probability $P(\mathcal{Y}|X, do(\mathcal{Z}'))$, where $\mathcal{Z}'$ is a subset of prior concepts $\mathcal{Z}$. To understand the nuanced effects of human intervention and reduce the computational demands during test-time intervention, we divide the prior concepts $\mathcal{Z} \subset \mathcal{N}$ into 25 equal segments $\{\mathcal{Z}_i^s\}_{i=1}^{25}$. Following these segments, we conduct 25 experiments by incrementally intervening the first $I$ segments $\mathcal{Z}' = \{\mathcal{Z}_i^s\}_{i=1}^{I}$ using the corresponding labels, and evaluate the accuracy of posterior concept on the ImageNet testing set. The intervention values are applied consistently with the intervention training method.

As shown in Figure 4, for the models that perform the intervention training, we observe a significant increase in the accuracy of PSM and DRL as the number of intervention nodes increases, reaching peak performance at approximately 95% top-1 accuracy. This highlights the effective human-AI interaction ability of NCG. Conversely, for the models without intervention training, we find that only the PSM model successfully learns the intervention rule on the CLIP model, indicating the challenge of developing reasoning ability from scratch without guidance. Additionally, we observe that the CBM model perform worse even when trained with the intervention method, demonstrating the importance of causal structure in reasoning. Interestingly, DRL + CLIP often improves performance with an increasing number of intervention nodes, although there is a noticeable drop in accuracy after a certain point (For details, please refer to Section A.8). This is because our NCG models, even without interventional training, are capable of learning causal dynamics and performing interventions inherently based on the underlying causal structure. Therefore, we believe the observed decline in accuracy for DRL + CLIP is an anticipated outcome.

In conclusion, this experiment demonstrates that our NCG method provides post-hoc interpretability, allowing users to understand how the reasoning process leads to specific predictions for posterior concepts. This aligns with existing research on interpretability in graph neural networks (Yuan et al., 2022). Additionally, our method extends beyond post-hoc analysis by enabling users to actively influence the model's reasoning during testing through test-time interventions.

## 5 CONCLUSIONS

We have developed a novel classification paradigm, neural causal graph, designed to facilitate interpretable and intervenable reasoning with enhanced generalization capabilities. NCG incorporates a structural causal model into the classification framework, enabling the disentanglement of complex knowledge representations and structures. This integration allows for precise reasoning based on concept information derived directly from an underlying causal graph, enhancing the model's ability to understand and interpret the data. Additionally, we developed an innovative intervention training and testing approach that enhances the NCG model's reasoning abilities and fosters collaboration between humans and AI. This approach involves posing "what if" questions, which challenge the model to consider intervention scenarios and adapt its predictions in the training stage, thereby enhancing its ability to handle interventions at test time. Finally, our experimental results demonstrate that the NCG approach has the potential to achieve interpretable and intervenable reasoning capabilities in classification tasks.

ACKNOWLEDGMENTS

This work is supported by the Natural Science Foundation of Hunan Province (No. 2022JJ30159 and No. 2023JJ20013), the National Natural Science Foundation of China (No. 61802121), and the Fundamental Research Funds for the Central Universities.

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

---

**Algorithm 1** Pseudo code for constructing the causal graph.

---

**Input:** dataset $\mathcal{D} = \{(X_1, y_1), (X_2, y_2), \ldots, (X_m, y_m)\}$, directed acyclic graph $\mathcal{G}^{WordNet}$ of Word-Net.
**Output:** causal graph $\mathcal{G} = \{\mathcal{N}, \mathcal{E}\}$ for the given dataset, multilabel $\tilde{\mathcal{Y}}$ of all examples, causal weight matrix $A$ for the causal graph.
    $\mathcal{G} \leftarrow$ Graph_Construction($\mathcal{D}$, $\mathcal{G}^{WordNet}$)
    $\tilde{\mathcal{Y}} = \{\tilde{Y}_1, \tilde{Y}_2, \ldots, \tilde{Y}_m\} \leftarrow$ Multilabel_Assignment($\mathcal{D}$, $\mathcal{G}$)
    **for** each edge $(C_i, C_j)$ in $\mathcal{G}$ **do**
        $A_{ij} \leftarrow$ Causal_Effect_Calculation($C_i, C_j, \{X_k, \tilde{Y}_k\}_1^m$)
    **end for**
    **return** $\mathcal{G}, \tilde{\mathcal{Y}}, A$

---

# A APPENDIX

This appendix is organized as follows:

- Section A.1 presents algorithms for constructing and training the NCG.

- Section A.2 offers an overview of the causal graph for the Bird dataset.

- Section A.3 describes the detailed implementation of propensity score matching (PSM) and doubly robust learning (DRL) for causal effect calculation in NCG.

- Section A.4 delves into the detailed reasoning process of NCG.

- Section A.5 illustrates the intervention process on specific nodes of the NCG with examples.

- Section A.6 examines the effect of different intervention rates during the training phase on model performance.

- Section A.7 details the test-time intervention experiment with human annotators to demonstrate interpretability and interactive capabilities of NCG.

- Section A.8 analyzes the performance drop in DRL+CLIP without intervention training and explains its causes.

- Section A.9 shows a case study that demonstrates the concept reasoning process of NCG.

- Section A.10 provides an explanation of the rationale behind using top-level prior concepts as shared labels, clarifying their role in maintaining implementation simplicity and facilitating information propagation within the causal graph.

## A.1 ALGORITHMS

This section provides two algorithms using pseudo code for detailing the construction of the causal graph and the training procedure for the NCG network. Detailed implementations of Algorithm 1 and Algorithm 2 can be found in the Method section of the main manuscript.

## A.2 DATASET VISUALIZATION

We provide the concept and structure overview of the Bird Dataset. Since the graph of the ImageNet dataset is huge, we only illustrate the graph of the Bird dataset as a schematic diagram. As depicted in Figure 5, the Bird Dataset contains a total of 25 nodes, with 16 nodes representing prior concepts and 9 nodes representing posterior concepts. Each posterior concept has a unique set of ancestors, comprised of several prior concepts that share common features with the posterior concepts.

## A.3 CAUSAL EFFECT CALCULATION: DETAILED IMPLEMENTATION

This section provides the implementation details of the Propensity Score Matching (PSM) (Rosenbaum & Rubin, 1983; Stuart, 2010) and Doubly Robust Learning (DRL) methods (Bang & Robins, 2005; Kang & Schafer, 2007; Cao et al., 2009) used to estimate causal effects for edges in the NCG.

---

**Algorithm 2** Pseudo code for training the NCG network.

---

**Input:** multilabel dataset $\tilde{\mathcal{D}} = \{X_k, \tilde{Y}_k\}_1^m$, causal graph $\mathcal{G} = \{\mathcal{N}, \mathcal{E}\}$ for the given dataset, pre-trained encoder $h(\cdot)$, intervention_rate $p$, confidence value $c$.
**Output:** concept proposer $g(\cdot)$, NCG network $f(\cdot)$, learnable scaled weights $\mathbf{w}$.
  optimizer = initialize_optimizer($[g, f]$)
  **for** randomly sample a batch $\mathcal{B} = \{X_k, \tilde{Y}_k\}_1^{bs}$ from $\tilde{\mathcal{D}}$ **do**
    $\mathcal{N}^{do} \leftarrow$ sample_intervention_nodes($\mathcal{N}, p$)
    $\mathbf{Y} \leftarrow$ multihot_encoding($\{\tilde{Y}_k\}_1^{bs}$)
    $\mathbf{C} \leftarrow$ Concept_Construction($\{X_k\}_k^{bs}$; $[g, h]$)
    **for** each node $C_j$ in topological_sort($\mathcal{G}$) **do**
      **if** $C_j$ in $\mathcal{N}^{do}$ **then**
        $\mathbf{C}'_j \leftarrow (2 \times \mathbf{Y}[:, j] - 1) \times c$
      **else**
        $\mathbf{C}'_j \leftarrow$ Concept_Reasoning($\mathbf{C}[:, j]$, $Pa(C_j)$)
      **end if**
      $\mathbf{C}_j \leftarrow \mathbf{C}'_j$ # note: $\mathbf{C}_j == \mathbf{C}[:, j]$
    **end for**
    loss $\leftarrow$ Loss_Calculation($\mathbf{C}'$, $\mathbf{w}$, $\mathbf{Y}$)
    optimizer.zero_grad()
    loss.backward()
    optimizer.step()
  **end for**

---

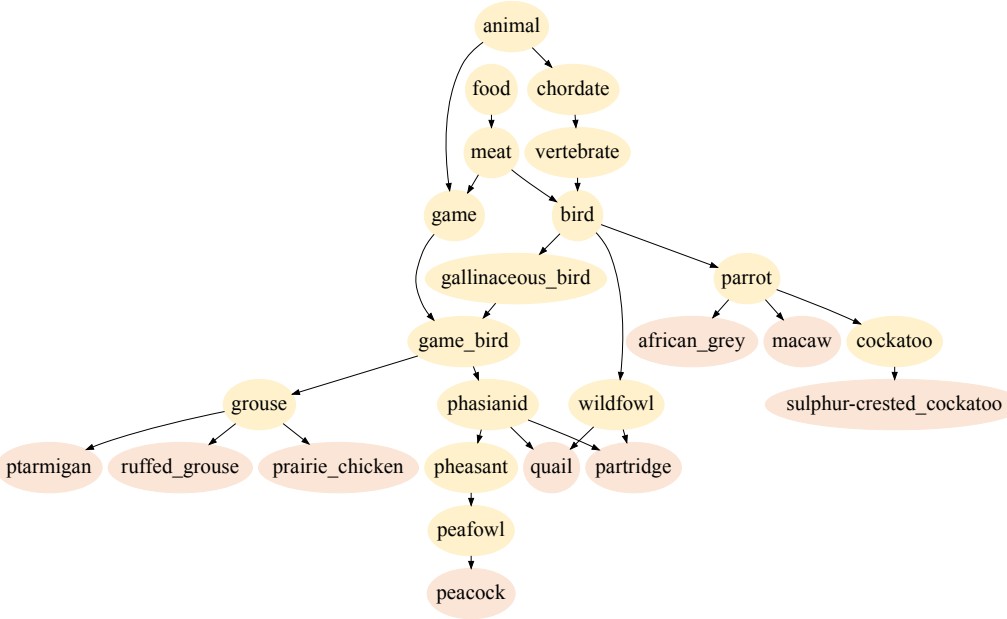

Figure 5: The illustration of concept structure of the Bird dataset.

### A.3.1 PROPENSITY SCORE MATCHING (PSM)

PSM estimates the causal effect of an edge in the NCG by comparing the outcomes of treatment and control groups with similar propensity scores. In our context, the source node of the edge represents the treatment $C$, while the target node represents the outcome $C'$. The steps are as follows:

1. **Propensity Score Estimation.** We model the propensity score (Rosenbaum & Rubin, 1983; Stuart, 2010), defined as $P(C|X)$ (the likelihood of treatment $C$ given background variables $X$), using logistic regression. Specifically, for a dataset of feature-concept pairs $\{(\mathbf{h}_i, C_i)\}_{i=1}^m$, where $\mathbf{h}_i \in \mathbb{R}^d$

represents the feature vector of input $X_i$ and $C_i \in \{0, 1\}$ is the treatment indicator, we train a logistic regression model with linear weight $\mathbf{w}$:

$$\log \frac{P(C = 1|X)}{1 - P(C = 1|X)} = \mathbf{w}^\top \mathbf{h}_i.$$

This yields the propensity score estimator $\hat{L}(X) = P(C = 1|X)$ for all samples.

2. **Matching.** Using the estimated propensity scores, we perform nearest neighbor matching (Rosenbaum & Rubin, 1983; Stuart, 2010) to pair each treated sample (C = 1) with a control sample (C = 0) that have the closest propensity scores. This minimizes the bias caused by confounding variables.

3. **Causal Effect Estimation.** The average treatment effect (ATE) is computed as the mean difference in outcomes between matched treatment and control groups:

$$ATE = \mathbb{E}[C'(C = 1) - C'(C = 0)], \tag{5}$$

where $C'(C = 1)$ and $C'(C = 0)$ are the potential outcomes under treatment and control conditions, respectively. These processes are calculated for all edges in the NCG.

### A.3.2 DOUBLY ROBUST LEARNING (DRL)

DRL provides a robust alternative to PSM by combining inverse probability weighting with regression-based outcome modeling. This ensures reliable estimation of causal effects, even when one of the models (propensity score or outcome) is misspecified. The steps are as follows:

1. **Outcome and Propensity Models.** DRL involves two predictive tasks: (i) estimating the propensity score $\hat{L}(X)$, as in PSM, and (ii) modeling the conditional outcome given the treatment and background variables, $\mathbb{E}[C'|X, C = 1]$ and $\mathbb{E}[C'|X, C = 0]$. We use logistic regression for both tasks to maintain simplicity and consistency:

$$\hat{Q}_c(X_i) = \mathbb{E}[C'|X, C = c] = \sigma(\mathbf{w}_c^\top \mathbf{h}_i), \quad c \in \{0, 1\},$$

where $\sigma(\cdot)$ is the sigmoid function.

2. **Treatment Effect Estimation.** The ATE is computed as:

$$\hat{ATE} = \frac{1}{m} \sum_{i=1}^{m} \left( \frac{C_i(C_i' - \hat{Q}_1(X_i))}{\hat{L}(X_i)} + \hat{Q}_1(X_i) \right)$$
$$- \frac{1}{m} \sum_{i=1}^{m} \left( \frac{(1 - C_i)(C_i' - \hat{Q}_0(X_i))}{1 - \hat{L}(X_i)} + \hat{Q}_0(X_i) \right), \tag{6}$$

where $\hat{Q}_1(X_i)$ and $\hat{Q}_0(X_i)$ are the predicted outcomes for treatment and control groups, respectively, and $\hat{L}(X_i)$ is the estimated propensity score. $m$ is the sample number of dataset.

### A.4 CONCEPT REASONING

In the Section of Concept Reasoning of the main paper, we discussed the reasoning function of NCG. As depicted in Figure 6, every node in NCG has been assigned an index that corresponds to the topological order. During the reasoning process, we apply the reasoning function on each node in order of their indexes, which ultimately produces a complete message propagation process.

### A.5 INTERVENTION ILLUSTRATION

In the Section of Intervention Training Method of the main paper, we discussed how NCG can be intervened to improve its reasoning capabilities. Specifically, as shown in Figure 7, we fix the concept logits of the 10th and 11th nodes (parrot and game bird), which cut off the influence from their ancestors. This intervention process can be thought of as a contextual prompt: if we assume that the bird in the image is more likely to be a parrot than a game bird, what exact type of bird could it possibly be? By answering such questions, NCG can better understand the causal relationships between different concepts and make more accurate predictions.

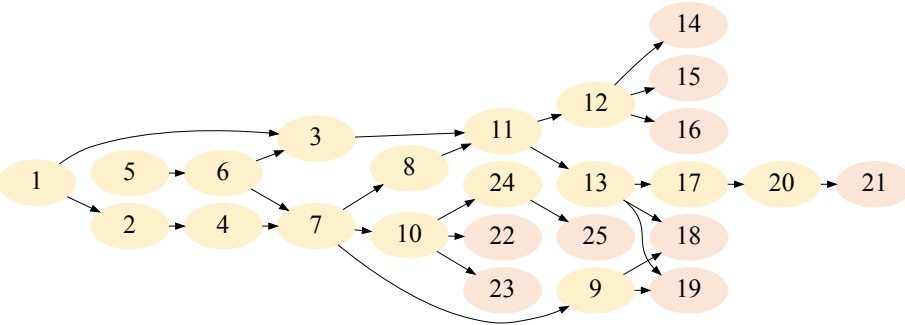

Figure 6: The nodes of NCG on the Bird dataset have been sorted topologically, with each node assigned an index according to this sorting.

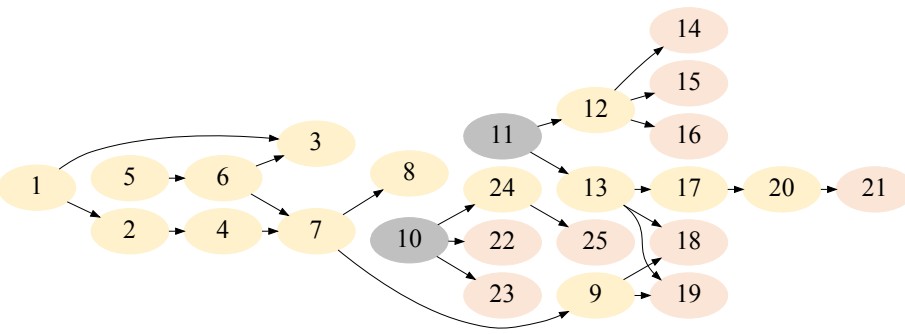

Figure 7: The intervenetion operation on two nodes of NCG.

## A.6 THE INTERVENTION RATE ON TRAINING TIME

Intervention training is a crucial technique in the NCG method, and it is necessary to study the impact of different intervention rates during the training phase. The results from Figure 8 indicate two important observations.

Firstly, we notice that the performance of PSM and DRL methods gradually increases with the increasing of intervention rate. The highest performance is observed at the 15% intervention rate, after which the performance starts to decline. This can be attributed to the fact that with too many prior concept nodes being fixed, the optimization process is being underly constrained, leading to poor generalization performance. Secondly, we find that when intervention training is applied with appropriate intervention rates, PSM and DRL methods show a consistent improvement in performance compared to the CBM method. This suggests that the NCG approach effectively captures the relationship between different concepts, while the CBM method fails to learn knowledge because it lacks of the concept structure that makes the intervention become noise.

## A.7 TEST-TIME INTERVENTION WITH HUMAN ANNOTATORS

To demonstrate the interpretability of our model, i.e., allowing humans to understand and participate in the reasoning process, we conducted a test-time intervention experiment involving real human annotators. The goal of this experiment is to evaluate whether human interventions could improve model predictions by leveraging the interpretable structure of the neural causal graph. Below, we detail the experimental setup, process, and results.

- **Sample Selection**: We selected 198 testing samples from the Bird dataset, covering 9 posterior concepts, to ensure diversity in the evaluation.
- **Intervention Process**:

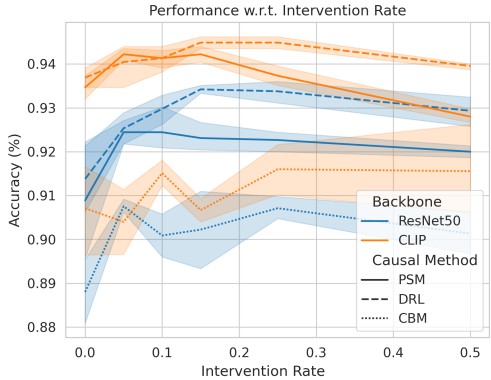

Figure 8: The performance of NCG with different intervention rate (0, 0.05, 0.10, 0.15, 0.25, 0.50) on the Bird dataset.

Table 5: Effect of Human Interventions on Model Accuracy. "H" is the abbreviation for "Human".

|  | Pre-Intervention | H1 | H2 | H3 | H4 | H5 | Human Avg. |
|---|---|---|---|---|---|---|---|
| **Correct Predictions** | 178 | 191 | 188 | 186 | 191 | 189 | 189 |
| **Accuracy (%)** | 89.9 | 96.5 | 94.9 | 93.9 | 96.5 | 95.5 | 95.5 |

1. Each image was classified by the NCG (PSM) model, which generated predicted probabilities for all concepts in the graph. These predictions were presented in a format similar to Figure 9 in the main paper.
2. Human annotators were shown the predicted probability graph alongside the original image. Annotators were asked to provide interventions by labeling one or more prior concepts as either 0 (absent) or 1 (present), reflecting their judgment of what the image represents.
3. The annotated prior concepts were used to update the graph, and the NCG (PSM) model performed reasoning based on the updated values and the input image.

- **Evaluation Metric**: The accuracy of the model's predictions before and after user intervention was recorded to measure the effectiveness of the human-guided updates.

We conducted this study with five human annotators who interacted with the model during the test-time intervention process. The results, summarized in Table 5, show a significant improvement in model accuracy after human interventions.

The results indicate that human interventions significantly improved the model's prediction accuracy, increasing it from 89.9% pre-intervention to an average of 95.5% across the five annotators. This demonstrates that the NCG framework enables humans to interact effectively with the model by interpreting and adjusting its causal reasoning process. The ability to intervene dynamically highlights the interpretability and flexibility of our approach, supporting sophisticated post-hoc analysis and facilitating human-AI collaboration.

This experiment validates the claim that our method yields interpretable concepts and arranges them in a manner that is understandable and actionable for human users.

A.8  PERFORMANCE DROP FOR DRL+CLIP W/O INTERVENTION TRAINING

As shown in Figure 4, the accuracy drop after around 1300 nodes likely arises because, without intervention training, the model can not reliably learn causal dependencies of graph, particularly as the number of nodes grows. There are three main reasons that, in an unexpected combination, led to this phenomenon when the model lacks interventional training. Note that our proposed NCG models with intervention training do not suffer from this issue, showing the necessity of our proposed intervention training process.

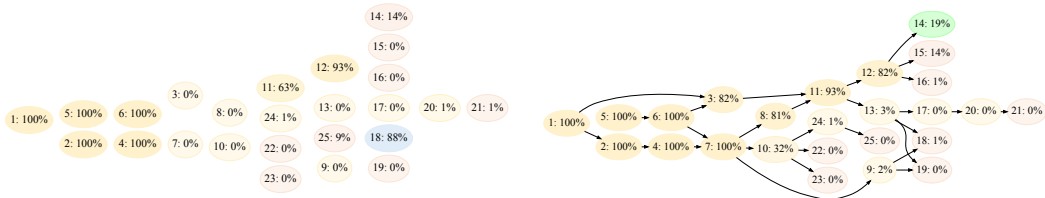

(a) Before graph reasoning. The 18th node with blue color represents the wrong prediction.

(b) After graph reasoning. The 14th node with green color indicates the correct prediction.

Figure 9: An example is selected from the Bird dataset to visualize the effectiveness of NCG. The first figure shows the sigmoid probabilities of the concept logits obtained from the concept proposer. The second figure illustrates the sigmoid probabilities of the updated concept logits after performing concept reasoning. In both figures, the yellow nodes represent the prior concepts of NCG, while the other color nodes represent the posterior concepts.

1. Positive Inductive Bias in the Causal Graph: The model inherently interprets positive logits in the formula $\mathbf{c}_j = \phi(g(h(X))_j + \sum_i A_{ij}\mathbf{c}_i + \mathbf{U}_{\mathbf{C}_j})$ as an indication of the presence of certain concepts in the images. This is a reasonable assumption given the structure of our causal graph, which has an inductive bias toward positive causal relationships. As detailed in the paper, the causal effect weights are typically estimated with 1 representing the presence of a concept and 0 representing its absence. This means that the estimated causal edges generally have positive weights, leading to an increase in the logits of child nodes when the logits of the corresponding parent concepts increase.

2. Negative Causal Weights in DRL Estimation: Although most of the estimated edge weights are positive, there are few exceptions where the causal weights are negative. This occurs because the DRL method uses a regression-based estimator to calculate causal effects, and in some cases, this results in a negative weight. When these negative weights exist, to increase the logits of their child nodes, the model must reduce the logits of parent concepts.

3. Intervention on the Last 100 Prior Concepts Close to Posterior Concepts: We observed that the last 100 prior concepts in the causal graph are most closely related to the posterior concepts within one-hop. During the intervention, we used values of -5 and 5 to manipulate the prior concepts. In extreme cases where the causal weights are negative, applying a -5 intervention to the prior concepts was mistakenly interpreted by the model as an increase in the corresponding posterior concept logits. This misinterpretation possibly leads to wrong classification results, thus causing a sudden and significant drop in accuracy.

These factors combined to produce the drastic drop in model performance after the initial 1300 effective node interventions. Note that this situation arises because the model did not perform intervention training, and the fact that the model was able to improve performance on the first 1300 nodes is already a noteworthy result. Conversely, models that have undergone proper intervention training do not suffer from this issue, demonstrating the effectiveness of our approach.

## A.9 CASE STUDY

As demonstrated in prior experiments, our model leverages NCG to enhance classification capabilities beyond the multi-class and multi-label baseline. To clearly illustrate this improvement, we analyze classification variations before and after concept reasoning with an NCG model, as depicted in Figure 9. Initially, the model incorrectly predicts the bird as "quail" (node 18: 88%). However, after applying concept reasoning, the prediction shifts to "ptarmigan" (node 14: 19%). This adjustment indicates that NCG effectively refines predictions by leveraging the interconnected concept information within the graph. On closer inspection of the probability of the prior concepts "grouse" (node 12: 93%) and "phasianid" (node 13: 0%) before concept reasoning, we can observe that this instance is more likely to be a "grouse" than a "phasianid". Considering ptarmigans belong to the grouse family, the accurate post-reasoning prediction is "ptarmigan". This case study demonstrates NCG's capacity to integrate all relevant concepts and causal relationships, offering a comprehensive understanding of the example and showcasing effective integration of neural networks with causal inference for interpretable and intervenable classification.

## A.10 Role of Top-Level Prior Concepts as Shared Labels

As explained in Section 3.2.2, WordNet's structure ensures that examples with different labels are assigned distinct prior concepts, benefiting the training process by capturing meaningful relationships within the graph. However, in the Bird dataset, the top-level concept "animal" is labeled as "1" for all examples because all birds inherently belong to the "animal" category. This uniformity provides limited discriminative information for distinguishing between classes, which raises questions about its utility in the classification process.

While this might seem counterintuitive, it reflects a design choice aimed at maintaining simplicity and consistency in the implementation described in Section 3.2.2. Importantly, only a very small number of top-level concepts, such as "animal" are shared by all examples in the dataset. Furthermore, assigning "1" to these concepts ensures that they can propagate information to subsequent nodes in the causal graph, aligning with the hierarchical structure of the real world. This approach allows the model to represent and utilize inherent structures without introducing redundancy or complex designs.

