# OpenReview forum: "Neural Causal Graph for Interpretable and Intervenable Classification"
_ICLR.cc/2025/Conference — ICLR 2025 Poster_

### Official Review · Reviewer_CUVg · 2024-10-21

**Soundness:** 3
**Presentation:** 2
**Contribution:** 3
**Rating:** 6
**Confidence:** 3

**Summary:**

The authors introduce Neural Causal Graph, a framework that builds a DAG over concepts from a neural network to improve performance. The DAG yields small but significant classification accuracy improvements on ImageNet and the CUB Birds dataset. The method and results seem compelling, but the writing of the paper is unclear.

**Strengths:**

- The paper studies an interesting problem of inducing a causal graph over a classification task
- The paper's results show significant improvements in classification accuracy on 2 datasets
- The introduced methodology seems to be generally useful, especially for sharing information between many classes

**Weaknesses:**

- The paper's writing is unclear, especially in the methods section.
- The paper overclaims about interpretability, e.g. "This capability not only supports sophisticated post-hoc interpretation but also enables dynamic human-AI interactions" -- it is unclear that the method yields interpretable concepts or arranges them in a manner that allows humans to understand them. More experiments (ideally with real human users) would help to show this
  - On a related note, the "intervention training method" (Secs 3.2.5 and 4.5) are especially unclear. Why would one expect to be able to intervene on a concept in a CBM baseline and maintain accuracy?

**Questions:**

- I assume the "Bird Dataset" is [CUB-200](https://www.vision.caltech.edu/datasets/cub_200_2011/)? The authors should make this clear in the paper
- Can the authors explain their test-time intervention training and results more clearly?

---

> ### Author Response · Authors · 2024-11-22
> **Author Rebuttal (I)**
>
> # To Reviewer CUVg (R4)
>
> We appreciate your careful review and acknowledgment of our approach’s potential. Thank you for highlighting key areas for improvement, particularly around clarity and interpretability claims.
>
> * **R4W1: The paper's writing is unclear, especially in the methods section.**
>
> We apologize for any difficulty in understanding our proposed method. We believe that the difficulty mainly comes from the background knowledge of causal-related techniques. Therefore, in the appendix of the revised manuscript, we will include more detailed explanations and relevant citations of various causal-related components to better describe our method, including causal intervention and estimiation, propensity score matching, doubly robust learning, and intervention training method.
>
> * **R4W2: The paper overclaims about interpretability, e.g. "This capability not only supports sophisticated post-hoc interpretation but also enables dynamic human-AI interactions" -- it is unclear that the method yields interpretable concepts or arranges them in a manner that allows humans to understand them. More experiments (ideally with real human users) would help to show this**
>
> Thank you for this valuable suggestion. We appreciate this suggestion. To further clarify interpretability that allows humans to understand them, **we conducted a test-time intervention experiment involving real human annotators**. Here is the experimental settings:
>
> 1.	**Sample Selection**: We selected 198 testing samples across 9 posterior concepts from the Bird dataset.
> 2.	**Intervention Process**:
> 	*	Each image was first classified by the NCG (PSM) model, which generated the predicted concept probabilities for the graph (similar to Figure 9 in the paper).
> 	*	Human annotators were then asked to reference both the predicted probability graph and the original image to provide interventions. Annotators are required to label at least one or more prior concepts by marking them as either 0 or 1, indicating what the image is or is not.
> 	*	These annotations were used to adjust the corresponding prior concept values, and the NCG (PSM) model then performed concept reasoning based on the image and user-provided interventions.
> 3.	**Evaluation**: The accuracy of the model’s predictions before and after user intervention was recorded.
>
> **To evaluate the effectiveness of human intervention, we conducted a study with five human annotators who interacted with the model during test-time interventions. The results, summarized in the table below, demonstrate that human intervention significantly improved model accuracy, increasing it from 89.9% (pre-intervention) to an average of 95.5% across annotators.**
>
> |  | pre-intervention | Human 1 | Human 2 | Human 3 | Human 4 | Human 5 | Human Average |
> |----------------------------|------------------|---------|---------|---------|---------|---------|---------------|
> | Correct Prediction Number | 178              | 191     | 188     | 186     | 191     | 189     | 189         |
> | Accuracy                  | 89.9%            | 96.5%   | 94.9%   | 93.9%   | 96.5%   | 95.5%   | 95.5%         |
>
> From the results, we can see that human interventions significantly improved model accuracy, demonstrating that humans can understand and interact with the reasoning process through the NCG framework. This demonstrates that our model provides interpretable and intervenable insights, enabling dynamic human-AI interactions.
>
> * **R4W3: On a related note, the "intervention training method" (Secs 3.2.5 and 4.5) are especially unclear. Why would one expect to be able to intervene on a concept in a CBM baseline and maintain accuracy?**
>
> Thank you for this question. Since it involves the motivation and methodology of our paper, we would like to address it in three parts:
>
> 1.	**Why test-time intervention?** Test-time intervention allows classification models to support user-AI interaction, enabling users to adjust predictions dynamically based on their domain knowledge.
>
> 2.	**Why intervention training?** Intervention training ensures that the model is prepared for test-time interventions by aligning the training and testing distributions. Without this method, the model may not handle interventions effectively. For example, in our framework, intervention training fixes specific prior concepts (e.g., logits set to +5 or -5) randomly during training, forcing the model to learn reasoning effectively under intervention conditions.
>
> 3.	**Why include CBM baselines?** Concept Bottleneck Models (CBM) also can perform interpretable and intervenable classification, which serves as a baseline for comparison. We expected that CBM would perform well on posterior concepts by correctly intervening its prior concepts when testing. Section 4.5 experiments were designed to highlight this comparison.
>
> We hope the clarifications provided will adequately address your concern.

---

> ### Author Response · Authors · 2024-11-22
> **Author Rebuttal (II)**
>
> * **R4Q1: I assume the "Bird Dataset" is CUB-200? The authors should make this clear in the paper.**
>
> Thank you for this clarification. The Bird dataset used in our experiments is a self-collected subset of ImageNet, containing 9 posterior concept labels, as described in L322–L323 and Table 1. For a more detailed overview, please refer to Section A.2 Dataset Visualization and Figure 5 in the appendix.
>
> * **R4Q2: Can the authors explain their test-time intervention training and results more clearly?**
>
> Thank you for raising this point. As described in Section 3.2.5 of the manuscript, we employ intervention training during the training phase to enable our model to perform test-time interventions effectively. Below is a concise explanation of the intervention training, test-time intervention, and experimental results:
>
> 1. **Intervention Training**: The intervention training method is designed to enhance the model’s reasoning ability by aligning the training and testing distributions, ensuring robustness under test-time interventions. Inspired by the principles of structural causal models (SCM), this approach simulates the effects of the $ do(\cdot) $ operator during training, isolating the influence of intervened nodes by eliminating their dependence on parent nodes. Specifically, during each training iteration, a subset of prior concepts is randomly selected with an intervention rate $ p = 0.15 $, and their values are fixed using the formula:
> $$
> Z_i = (2Y_i - 1) \times v,
> $$
> where $ Z_i $ is the intervened node value, $ Y_i \in \{0, 1\} $ represents the corresponding multi-label value, and $ v $ is an empirically determined confidence value set to $ 5 $. The sigmoid output of $ -5 $ or $ 5 $ reflects high certainty about the node’s intervention status. This process is designed to enable the model to address "What if" questions by learning the dynamics among concepts through intervention. By simulating test-time scenarios, the model is conditioned to predict posterior concepts based on both inferred and intervened prior concepts. An overview of the procedure for intervention training is detailed in Algorithm 2 (Appendix).
>
> 1. **Test-time Intervention**: After models trained with intervention training method, we propose a test-time intervention experiment to explore how human interaction with prior concepts influences the model’s performance. First, prior concepts have been arranged in topological order as described in Sections 3.1.1. Prior concepts (Z) are divided into 25 segments, and interventions on these segements are applied incrementally, enabling us to observe changes in posterior concept accuracy. The intervention values follow the same process as the intervention training phase (+5 / -5).
>
> 2. **Results**: Figure 4 shows that models trained with intervention methods achieve significant accuracy improvements (e.g., up to 95% top-1 accuracy). Models without intervention training, however, struggle to reason effectively under intervention scenarios. CBM performance demonstrates the importance of causal structures in reasoning, while our NCG results highlight the robust interaction capabilities under test-time interventions.
>
> We hope this explanation clarifies the training and testing dynamics. If additional clarification is needed, we would be happy to provide it. Thanks.

---

> ### Comment · Reviewer_CUVg · 2024-11-25
> **Thank you for your reply**
>
> I thank the author's for their comments and encourage them to include the added details in the manuscript. I have updated my score from 5 to 6.

---

> > ### Author Response · Authors · 2024-11-25
> > **Official Comment by Authors**
> >
> > Dear Reviewer CUVg,
> >
> > We truly appreciate your encouraging and insightful comments and suggestions. We will ensure that the additional details are thoroughly incorporated into the revised manuscript. Your evaluation is invaluable in helping us further refine and improve our work. Thank you once again for your time and support!
> >
> > Best regards,
> >
> > Authors

---

### Official Review · Reviewer_ytQg · 2024-10-30

**Soundness:** 3
**Presentation:** 2
**Contribution:** 3
**Rating:** 6
**Confidence:** 2

**Summary:**

The paper proposes a classification framework that integrates causal inference within neural networks to enhance interpretability and facilitate human-guided interventions. Experiments are conducted to see how performance compares against baselines such as concept-bottleneck models, then the effect of causal weights, ablation studies, and test time intervention capability.

Overall, the results seem to show that the performance is promising and test time intervention feasible.

**Strengths:**

* Integrating causal reasoning into deep neural networks is a nice feature of the paper, and not something (to the best of my knowledge) which as been explored before convincingly.
* The method appears quite general, using WordNet should generalize to most domains.
* Test time intervention should be useful in a HCI context, in more complex decision making which is higher stakes.

**Weaknesses:**

* First and foremost I found the paper pretty difficult to grasp exactly what had been done, a lot of it is heavily formalized without any intuitive explanations to facility easy reading, this is really going to limit who can read and build upon the work. Those who work in causal reasoning are mostly used to SCMs and it is pretty hard to understand the core differences this presents to that framework, I would really suggest re-writing the paper drastically. For example, Figure 1 is not even reference until page 4, but it's on page 2, this all makes the paper hard to follow.
* The reliance on WordNet, while smart, limits the method a lot because not all concepts/relations will be in this dataset.
* The test time intervention relies on using labels, but what if the human involved doesn't know the labels? The accuracy won't be as good as you are reporting.

**Questions:**

* Why did the regular classifiers in Table 2 perform worse than your method? This doesn't make sense to me as you are incorporating more constraints, did the causal reasoning actually result in better results?
* Did you add multi-lables for each datapoint by using WordNet? If so, then all images of e.g. Bird1 have all the same concept labels in every example you train on? Isn't that just making the classification problem more convoluted? Surely each image of Bird1 should have varying concept labels to truly benefit the training process and your method? Otherwise it's just the same classification problem, just with more more labels.
* Can you make explicit how you think test time intervention could be used in application?
* Can you run your method NCG without the causal structure to understand how much benefit is attributed to the causal graph itself?

---

> ### Author Response · Authors · 2024-11-22
> **Author Rebuttal (I)**
>
> # To Reviewer ytQg (R3)
>
> Thank you for your thoughtful review and positive feedback on our integration of causal reasoning into neural networks. Below, we provide detailed point-by-point responses to your comments.
>
> * **R3W1: First and foremost I found the paper pretty difficult to grasp exactly what had been done, a lot of it is heavily formalized without any intuitive explanations to facility easy reading, this is really going to limit who can read and build upon the work. ... I would really suggest re-writing the paper ...**
>
> We apologize for any difficulty in understanding our proposed method. To improve clarity, we are considering moving the formulaic details to the Appendix, which could make the paper more accessible to readers without extensive SCM research experience.
>
> **Regarding the structure of our writing, we deliberated between two approaches:**
>
> 1.	**A traditional neural network innovation-focused structure (e.g., GNNs, CBM)**. While this approach might seem familiar to readers, it would fail to fully justify why SCM components (e.g., causal intervention, PSM, DRL, intervention training) are critical for enabling test-time intervention.
>
> 2.	**A structure aligned with causality literature, as currently adopted in our paper**. Our method follows the logical progression consistent with theoretical causal literature [1] ("Introduction to Probabilities, Graphs, and Causal Models" -> "Causal Diagrams and the Identification of Causal Effects" -> "Actions, Plans, and Direct Effects,"). This provides a coherent and rigorous foundation for combining the NCG paradigm with structural causal models.
>
> **We hope our approach strikes a balance between clarity and rigor while presenting the novelty of our method.** Thank you again for your constructive suggestions.
>
> [1] Causality: Models, Reasoning, and Inference. Judea Pearl
>
> * **R3W2: The reliance on WordNet, while smart, limits the method a lot because not all concepts/relations will be in this dataset.**
>
> Thank you for your insightful comments. **Since the additional causal graph enhances performance and facilitates Human-AI interaction, we consider it a feature rather than a limitation. For classification tasks where related concepts may not be available in WordNet, we can follow the Graph Construction process (Section 3.1.1) using alternative graph resources.** For example, by manually constructing a causal graph based on expert knowledge, we can integrate domain-specific insights and enable dynamic interventions during the inference process, which is a significant advantage. Additionally, we plan to explore the possibility of automatically constructing causal graphs using causal discovery techniques in future work. This advancement would allow our framework to adapt to a broader range of datasets with unknown or diverse relationships, further extending its applicability.
>
> * **R3W3: The test time intervention relies on using labels, but what if the human involved doesn't know the labels? The accuracy won't be as good as you are reporting.**
>
> Thank you for raising this important point. Our current research focuses on achieving better accuracy with correct human interventions, which allows for human-AI interaction and aligns with current research like CBM. Extending the NCG paradigm to handle noisy or uncertain interventions is an promising direction for future work. This would involve designing robust methods to mitigate the impact of imperfect interventions while maintaining performance.
>
> * **R3Q1: Why did the regular classifiers in Table 2 perform worse than your method? This doesn't make sense to me as you are incorporating more constraints, did the causal reasoning actually result in better results?**
>
> We appreciate your insightful comments. In fact, the improved performance of NCG models aligns with our hypothesis, as the causal graph introduces structured information that enhances inference capabilities. Specifically, we incorporated key components based on structural causal model theory, including the causal graph, causal effect calculation, and intervention training. These components not only enable effective test-time interventions but also contribute to improved overall model performance.
>
> In contrast to regular classifiers, which treat posterior concepts as independent, the NCG method utilizes causal relationships among concepts to improve reasoning. This structured approach enhances the model’s ability to infer and predict accurately. **Our experimental results (Tables 2, 3, and 4) clearly show that the NCG method consistently outperforms regular classifiers, largely due to these causal-based designs.**

---

> ### Author Response · Authors · 2024-11-22
> **Author Rebuttal (II)**
>
> * **R3Q2: Did you add multi-lables for each datapoint by using WordNet? If so, then all images of e.g. Bird1 have all the same concept labels in every example you train on? Isn't that just making the classification problem more convoluted? Surely each image of Bird1 should have varying concept labels to truly benefit the training process and your method? Otherwise it's just the same classification problem, just with more more labels.**
>
> Thank you for this question. Do you mean that the top-level prior concept in the causal graph is shared as a concept label by all examples (e.g., "animal" in the Bird dataset as shown in Figure 5)? **We believe it is better to label it as "1" than not because all examples in the Bird dataset actually belong to the concept "animal" in the real world, and thus makes it can correctly influences the subsequent nodes in the graph.**
>
> Moreover, as you mentioned, we assign multi-labels to each example using WordNet as explained in Section 3.1.2. Examples with different labels will take their corresponding ancestor nodes as prior concept labels, resulting in different sets of concept labels and can really benefit the NCG training process. Conversely, examples with the same labels will share the same prior concept labels. This approach enables the model to learn additional information about the relationships between prior and posterior concepts, rather than simply adding redundant labels. We hope these explanations could address your concerns.
>
> * **R3Q3: Can you make explicit how you think test time intervention could be used in application?**
>
> Thank you for your insightful question. We believe that test-time intervention could be particularly valuable in applications requiring human-AI interaction. **For instance, beyond providing classification results, users may also need post-hoc interpretability to understand why a certain prediction was made. Moreover, users want to leverage their own expert knowledge to intervene in the intermediate reasoning process, enabling the model to generate more accurate and reasonable outputs.** For example, a potential application is in medical diagnosis based on patient records, where complex causal relationships between symptoms and diseases inform predictions. It is often challenging for doctors to notice and accurately identify all relevant symptoms and diseases. Our NCG method have potential to facilitate interaction between doctors and the model, allowing predictions about symptoms and diseases while deepening doctors’ understanding of patients. To improve diagnostic accuracy, doctors could inject their domain knowledge into the model during the intervention process to refine its reasoning.
>
> **To further illustrate the application of test-time intervention, we conducted an experiment involving real human annotators, as detailed in our response to R4W2. This experiment demonstrates that humans can effectively understand and interact with the reasoning process through the NCG framework. We hope this addresses your concerns about the practical application of our method.**
>
> For more details on the technical implementation and meaning of test-time interventions, please refer to Appendix A.4.1 (Intervention Illustration) and Figure 7 in our paper.
>
> Thank you again for your thoughtful question!
>
> * **R3Q4: Can you run your method NCG without the causal structure to understand how much benefit is attributed to the causal graph itself?**
>
> Thank you for this important suggestion. Understanding the contribution of the causal graph is indeed crucial, and we appreciate your emphasis on this point. In our manuscript, we have evaluated two variants of our model without the causal structure to examine its impact:
>
> 1.	**Multi-label model**: As described in Section 4.2 and Table 2, this variant uses multi-labels (prior and posterior concepts) but lacks the causal graph. It performs worse than the full NCG model, indicating the importance of the causal graph for improved performance.
> 2.	**Zero Weight model**: In Section 4.3 and Table 3, this variant disables causal weights, preventing concept nodes from propagating information. The performance of this “Zero Weight” model is also worse than NCG with PSM or DRL weights.
>
> **These results support our conclusion that the causal structure provides supplemental information for reasoning, enabling the model to achieve better performance.**

---

> > ### Comment · Reviewer_ytQg · 2024-11-26
> >
> > *We apologize for any difficulty in understanding our proposed method. To improve clarity, we are considering moving the formulaic details to the Appendix, which could make the paper more accessible to readers without extensive SCM research experience.*
> >
> > Sorry, but saying you are “considering” to do this is not convincing, given you have around 2 weeks of rebuttal period, and the enormous effort other authors put into revision of the manuscripts, I would have expected you to have done this by now.
> >
> > ***
> >
> > *Thank you for your insightful comments. Since the additional causal graph enhances performance and facilitates Human-AI interaction, we consider it a feature rather than a limitation. For classification tasks where related concepts may not be available in WordNet, we can follow the Graph Construction process (Section 3.1.1) using alternative graph resources. For example, by manually constructing a causal graph based on expert knowledge, we can integrate domain-specific insights and enable dynamic interventions during the inference process, which is a significant advantage.*
> >
> > Does this mean that without wordnet, all causal graphs with your method need to be hand-defined?
> >
> > ***
> >
> > *Thank you for raising this important point. Our current research focuses on achieving better accuracy with correct human interventions, which allows for human-AI interaction and aligns with current research like CBM. Extending the NCG paradigm to handle noisy or uncertain interventions is a promising direction for future work. This would involve designing robust methods to mitigate the impact of imperfect interventions while maintaining performance.*
> >
> > Ok, so test time intervention assumes the expert knows the labels already? That's totally fine, but I need the authors to give me a crystal clear application where this would be useful, and preferably demonstrate this in the paper convincingly.
> >
> > ***
> >
> > *Thank you for this question. Do you mean that the top-level prior concept in the causal graph is shared as a concept label by all examples (e.g., "animal" in the Bird dataset as shown in Figure 5)? We believe it is better to label it as "1" than not because all examples in the Bird dataset actually belong to the concept "animal" in the real world, and thus makes it can correctly influences the subsequent nodes in the graph.
> > Moreover, as you mentioned, we assign multi-labels to each example using WordNet as explained in Section 3.1.2. Examples with different labels will take their corresponding ancestor nodes as prior concept labels, resulting in different sets of concept labels and can really benefit the NCG training process. Conversely, examples with the same labels will share the same prior concept labels. This approach enables the model to learn additional information about the relationships between prior and posterior concepts, rather than simply adding redundant labels. We hope these explanations could address your concerns.*
> >
> > I really think all this should be clearer, it ties back to the issue of presentation which other reviewers have also mentioned.
> >
> > ***
> > *R3Q3: Can you make explicit how you think test time intervention could be used in application? Thank you for your insightful question. We believe that test-time intervention could be particularly valuable in applications requiring human-AI interaction.*
> >
> >
> > Thanks for all your work here, I read the experiment you posted to the other reviewer. To my understanding, you are basically getting humans to correct the model’s output. I don’t see the point of this. The problem I have is that you are saying “We --believe-- that test-time intervention –could-- be.. valuable”. The important words being “believe” and “could”. The authors have not demonstrated this in a convincing way. Moreover, there are only 5 users in this study, which I assume are not randomly sampled from the population right? This not convincing at all.
> >
> > ***
> >
> > *Thank you for this important suggestion. Understanding the contribution of the causal graph is indeed crucial, and we appreciate your emphasis on this point. In our manuscript, we have evaluated two variants of our model without the causal structure to examine its impact:*
> >
> > Thanks for this! Overall, I think there's still too many issues with the paper for me to increase the score currently. If the authors could address the remaining points, I can reconsider this.

---

> > > ### Author Response · Authors · 2024-11-28
> > > **Author Rebuttal (III)**
> > >
> > > Thank you for your thorough review and valuable feedback, which have been instrumental in improving the quality and clarity of our work.
> > >
> > > * **R3W1: Sorry, but saying you are “considering” to do this is not convincing, given you have around 2 weeks of rebuttal period, and the enormous effort other authors put into revision of the manuscripts, I would have expected you to have done this by now.**
> > >
> > > Thank you for emphasizing the importance of making concrete revisions. It took me a bit longer than expected to gather all the necessary details. We have considered your concern by revising the manuscript and moving the formulaic details to the Appendix, to improve the readability for readers unfamiliar with SCMs. Specifically, we have revised the following:
> > >
> > > 1.	We moved the detailed formulas for Causal Effect Calculation (PSM and DRL) to Appendix A.3, as suggested. We then expanded the Appendix to include a thorough discussion of PSM and DRL, providing technical depth while keeping the main sections focused.
> > >
> > > 2.	To make the paper more accessible, we have added a dedicated Preliminaries section, introducing the basic concepts and notation of SCMs.
> > >
> > > These revisions have been clearly highlighted in blue in the revised manuscript. We are very grateful for your suggestion, which has significantly enhanced the clarity and accessibility of our paper.
> > >
> > > * **R3W2: Does this mean that without WordNet, all causal graphs with your method need to be hand-defined?**
> > >
> > > Thank you for this constructive question. As we discussed in the R3W2 of Author Rebuttal (I), we would like to clarify that while manually defining causal graphs based on expert knowledge is one feasible approach, it is not the only option. **The construction of causal graphs with our method depends on the specific application domain and the availability of resources.** For examples:
> > >
> > > 1.	In image classification domain, as demonstrated in our paper, WordNet was leveraged to construct the causal graph for ImageNet. WordNet’s rich taxonomy with over 155,000 words and relationships, making it an ideal resource for constructing domain-specific causal graphs.
> > >
> > > 2.	In other fields like medical applications, existing graph databases, such as KEGG (Kyoto Encyclopedia of Genes and Genomes), can be used to build causal graphs representing relationships between diseases, drugs, and pathogens.
> > >
> > > 3.	Beyond pre-existing resources, automated methods such as causal discovery [1] and causal representation learning [2] algorithms can also be applied to generate causal graphs from data in domains lacking structured knowledge bases.
> > >
> > > **Note that we emphasize that our paper’s main contribution lies in proposing the neural causal graph for interpretable and intervenable classification.** While we evaluated our method using WordNet and ImageNet, applying our approach to construct causal graphs in other domains remains a broader and open research question. Your question has highlighted the potential for further exploration in this area, and we sincerely appreciate your thoughtful comments.
> > >
> > > [1] A Survey on Causal Discovery: Theory and Practice
> > >
> > > [2] Toward Causal Representation Learning

---

> > > ### Author Response · Authors · 2024-11-28
> > > **Author Rebuttal (IV)**
> > >
> > > * **R3W3: Ok, so test-time intervention assumes the expert knows the labels already? That’s totally fine, but I need the authors to give me a crystal-clear application where this would be useful, and preferably demonstrate this in the paper convincingly.**
> > >
> > > Thank you for understanding the standard setup of test-time intervention experiment, which is based on the widely observed fact that humans generally find it easier to identify general prior concepts (e.g. parrot, grouse, cough, chest pain) than specific posterior concepts (e.g. ruffed_grouse, macaw, bronchitis, pneumonia). **We appreciate your comments on applications, but we would like to clarify that the main focus of our paper is on proposing the NCG framework for interpretable and intervenable classification (Image Classification), rather than demonstrating broader applications.**
> > >
> > > To address your concerns, we further elaborate on the applicability of test-time intervention technique from three perspectives to clarify its value:
> > > 1. **Practical Application in Image Classification.** We conducted extensive experiments in Section 4.5 Test-time Intervention, where we demonstrated that by enabling intervention on prior concepts, the NCG framework achieves nearly 95% top-1 accuracy on the posterior concepts of ImageNet. This result highlights that NCG **supports interactive intervention during testing --- a capability absent in traditional multi-label classification models**.
> > >
> > > 2. **Validated Application with Real Human Interaction.** Additionally, as shown in R4W2, we performed a user study involving five human annotators to further validate the practical utility of test-time intervention. The results showed that human interventions improved model predictions, **demonstrating the usefulness of the framework for incorporating human knowledge during the interaction process.**
> > >
> > > 3. **Potential Application in Medical Diagnosis.**
> > >    1. **While our main focus is to propose the neural causal graph framework for interpretable and intervenable classification, we are happy to further discuss possible applications to demonstrate the value of test-time intervention based on your comments.**
> > >    2. As previously discussed, one potential application of test-time intervention is in medical diagnosis based on patient records, where complex causal relationships between symptoms and diseases inform predictions. For example, a doctor might observe that a patient exhibits symptoms such as persistent cough and fever but initially overlooks the subtle signs of chest pain. Using the NCG framework, the model could provide an initial prediction, suggesting potential diseases like bronchitis or pneumonia based on the observed symptoms. If the doctor realizes the chest pain is a critical symptom that should influence the diagnosis, they could intervene by explicitly marking this symptom as present in the model. The NCG would then update its reasoning process, propagating the causal effects of the new input across the graph. This could refine the prediction to consider diseases such as pleurisy or pulmonary embolism, which are more consistent with the full set of symptoms.
> > >    3. **Through this interaction, the model not only helps the doctor consider diseases they may have overlooked but also enables the doctor to inject their domain knowledge into the reasoning process, deepening their understanding of the patient’s condition and improving diagnostic accuracy.**
> > >
> > > We hope these validated examples, along with the potential use case, clearly demonstrate the practical applications and broader potential of test-time intervention. We sincerely appreciate your encouragement to delve deeper into this topic and thank you again for your insightful suggestion.

---

> > > ### Author Response · Authors · 2024-11-28
> > > **Author Rebuttal (V)**
> > >
> > > * **R3Q2: I really think all this should be clearer, it ties back to the issue of presentation which other reviewers have also mentioned.**
> > >
> > > Thank you for your thoughtful feedback. To address your concern regarding the specific reasoning of "top-level prior concepts shared as concept labels by all examples", we have clarified this in the revised manuscript. For improved clarity, the explanation has been moved to the Appendix, where we state:
> > > > As explained in Section 3.2.2, WordNet’s structure ensures that examples with different labels are assigned distinct prior concepts, benefiting the training process by capturing meaningful relationships within the graph. However, in the Bird dataset, the top-level concept "animal" is labeled as "1" for all examples because all birds inherently belong to the "animal" category. This uniformity provides limited discriminative information for distinguishing between classes, which raises questions about its utility in the classification process.
> > >
> > > > While this might seem counterintuitive, it reflects a design choice aimed at maintaining simplicity and consistency in the implementation described in Section 3.2.2. Importantly, only a very small number of top-level concepts, such as "animal" are shared by all examples in the dataset. Furthermore, assigning "1" to these concepts ensures that they can propagate information to subsequent nodes in the causal graph, aligning with the hierarchical structure of the real world. This approach allows the model to represent and utilize inherent structures without introducing redundancy or complex designs.
> > >
> > > We sincerely appreciate your suggestion and thank you for encouraging us to clarify the reasoning behind our implementation.
> > >
> > > * **R3Q3: Thanks for all your work here, I read the experiment you posted to the other reviewer. To my understanding, you are basically getting humans to correct the model’s output. I don’t see the point of this. The problem I have is that you are saying “We --believe-- that test-time intervention –could-- be.. valuable”. The important words being “believe” and “could”. The authors have not demonstrated this in a convincing way. Moreover, there are only 5 users in this study, which I assume are not randomly sampled from the population right? This not convincing at all.**
> > >
> > > Thank you for raising this important point and providing detailed feedback on our test-time intervention settings. **Considering your multiple concerns about this technique, we aim to address them from several perspectives, including the research motivation, the intervention process, experimental validation, and its connection to human-AI interaction:**
> > >
> > > 1.	**Research Motivation and Novelty.** The neural causal graph framework is specifically designed to address the need for **interpretable and intervenable classification**, with test-time intervention being a novel feature that most traditional multi-label classification neural networks do not support.
> > >
> > > 2. **Clarification of the Intervention Process.** The setup of test-time intervention is based on the observation that humans can more easily identify **prior concepts** (e.g., "parrot," "grouse," "cough," "chest pain") compared to the more specific **posterior concepts** (e.g., "macaw," "ruffed_grouse," "bronchitis," "pneumonia"). This approach leverages the hierarchical causal structure of the NCG framework, where prior concepts act as interpretable intermediate nodes that propagate influence to more detailed predictions.
> > >
> > > 3. **Convincing Experimental Validation.** In Section 4.5, we demonstrated that NCG achieves **nearly 95% top-1 accuracy** on ImageNet through test-time intervention, highlighting its effectiveness in leveraging prior concepts during inference. Models lacking intervention training or causal graph cannot achieve this capability, underscoring the effectiveness of the proposed framework.
> > >
> > > 4. **Human Interaction Study.** Although the human intervention study involved only 5 users, its main purpose was to validate that the model can understand and incorporate human knowledge during inference. We respectfully note that none of the 5 annotators had expertise or prior specialized knowledge in bird-related domains. **Furthermore, a successful intervention relies not only on human knowledge but also on the model’s ability to correctly reason based on the intervention. The positive experimental results clearly demonstrate the combined effect of human input and the model’s ability to reason based on interventions. This validates the effectiveness of the model’s reasoning capability when guided by interventions, which directly validates the research motivation of our work.**
> > >
> > > We sincerely appreciate your thoughtful comments, which have prompted us to further clarify the motivation, design, and validation of test-time intervention. We hope this response addresses your concerns and highlights the value of this feature within the scope of our research.

---

> > > ### Author Response · Authors · 2024-11-28
> > > **Author Rebuttal (VI)**
> > >
> > > * **R3Q4: Thanks for this! Overall, I think there's still too many issues with the paper for me to increase the score currently. If the authors could address the remaining points, I can reconsider this.**
> > >
> > > Thank you for acknowledging our clarification on R3Q4. We truly appreciate your engagement and constructive feedback, which have been instrumental in improving our paper. We have carefully addressed your concerns and made efforts to enhance the clarity in the revised manuscript. We hope these updates resolve your concerns, and we remain open to further discussion to ensure our contributions are clearly communicated. Thank you again for your thoughtful comments!

---

> > > > ### Comment · Reviewer_ytQg · 2024-12-02
> > > >
> > > > I thank the authors for their rebuttal and addressing all my points.
> > > >
> > > > Overall, most of my concerns I don't feel were well addressed honestly, such as clear applications that test time intervention would be useful for, the authors are mostly doubling down on the fact the paper is not concerned with this but rather just developing a theoretical method that might be useful in the future for something (which is one of the biggest flaws of XAI research). Moreover, the new experiment with 5 users is difficult to interpret, 5 users is really just a pilot study, but might provide a somewhat useful datapoint for future research.
> > > >
> > > > However, the authors have made some earnest attempts to improve the work, and overall it has one or two nice ideas worth publishing I feel, so I appreciate that. I would not strongly suggest accept or reject, but I will increase the score from 5 -> 6 to reflect their effort, but lower my confidence unfortunately.

---

> > > > > ### Author Response · Authors · 2024-12-03
> > > > > **Official Comment by Authors**
> > > > >
> > > > > Dear Reviewer ytQg,
> > > > >
> > > > > Thank you for your thoughtful follow-up and for taking the time to carefully review our rebuttal. We sincerely appreciate your detailed feedback and the increased score, which reflects your recognition of our efforts to improve the manuscript.
> > > > >
> > > > > Our primary goal in this work was to focus on developing the NCG framework for interpretable and intervenable classification, with test-time intervention as a novel feature that we believe holds potential for real-world applications.
> > > > >
> > > > > Your acknowledgment of the strengths in our methodology, particularly the innovative ideas introduced, means a great deal to us. Your feedback and increased score motivate us to continue refining and expanding on this work to better demonstrate its utility and impact.
> > > > >
> > > > > Thank you again for your engagement, constructive comments, and encouragement. We are grateful for the opportunity to improve our work based on your valuable suggestions.
> > > > >
> > > > > Best regards,
> > > > >
> > > > > Authors

---

### Official Review · Reviewer_sJC6 · 2024-10-31

**Soundness:** 2
**Presentation:** 1
**Contribution:** 2
**Rating:** 5
**Confidence:** 3

**Summary:**

This paper proposes a new interpretable and intervenable neural network. The main idea of the proposed models is to incorporate a causal graph among latent concepts and target labels into a neural network. To train such a model, the authors propose calculating the causal effects between concepts and labels by propensity score matching and doubly robust learning. Its prediction of each concept or label is modeled by the linear combination of the estimated causal effects of the parent concepts in a causal graph, as well as the output of the encoder in a pre-training network. The authors also propose an intervention training technique that generates counterfactual samples by intervening in selected training samples according to the causal graph. Experimental results demonstrated that the proposed method attained higher accuracy than the existing baselines, including the concept bottleneck model.

**Strengths:**

- S1. The authors proposed a new concept bottleneck model by incorporating a causal graph that represents causal relationships between latent concepts and target labels. I think it is interesting and useful to generate concepts corresponding to the labels and their causal directions using WordNet.
- S2. The proposed method slightly improved the accuracy and F1 score on the Bird and ImageNet datasets compared to the baselines.

**Weaknesses:**

- W1. It was difficult for me to understand the whole picture of the proposed method. Maybe this is because this paper is structured to describe the proposed method in a bottom-up manner. I believe describing the proposed method in a top-down manner, that is, first giving an abstracted model of the proposed model and then describing its formulation of each detail, may help readers understand the proposed method. In addition, while the authors introduce several unique techniques (e.g., causal effect calculation, concept reasoning, and intervention training), it was unclear to me what purpose these techniques are introduced for and what observation or motivation they are based on. If my understanding is correct, the proposed method can be regarded as an extension of the concept bottleneck models incorporating causal relationships between labels and concepts. I think this paper may be improved by organizing it following the original paper on the concept bottleneck models [Koh et al. 2020].
- W2. This paper claims that the proposed method is interpretable. However, I could not find any discussion or evidence demonstrating the interpretability of the proposed method.
- W3. I found some confusing notational errors. For example, while $n$ represents the total number of labels in Line 153, it denotes the total number of nodes in a causal graph in Line 268. Also, in Equation (5), $n$ appears to represent the total number of training samples. Furthermore, while $C$ is defined as a binary variable in Line 203, it denotes the set of concept variables in Line 275. In addition, I think $w_{ij}$ in Equation (6) is correctly $A_{ij}$. Since such notational errors prevent readers from understanding the proposed method, I believe the mathematical part of this paper should be corrected entirely.

**Questions:**

- Q1. Concerning W1, is my understanding of the proposed method correct?
- Q2. Concerning W2, from what perspective do the authors claim that the proposed model is interpretable?
- Q3. What was the purpose of the experiments in Section 4.5? If the test samples were intervened and their concepts and labels were perturbed according to a causal graph, it seems natural that the accuracy of the proposed method increased and that of CBM decreased since the former was trained with the causal graph and the latter did not. Can this experiment be considered fair?

---

> ### Author Response · Authors · 2024-11-22
> **Author Rebuttal (I)**
>
> # To Reviewer sJC6 (R2)
>
> Thank you for your thorough review and constructive feedback. We acknowledge the areas you highlighted for improvement and are committed to addressing them in our revised manuscript to improve readability and understanding.
>
> * **R2W1 & R2Q1: The bottom-up presentation was challenging to follow; a top-down approach may make it clearer. Several unique techniques (e.g., causal effect calculation, concept reasoning, and intervention training) were unclear to me. I think this paper may be improved by organizing it following the original paper on the concept bottleneck models [Koh et al. 2020]. Is my understanding of the proposed method correct?**
>
> We apologize for any difficulty in understanding our proposed method and appreciate your thoughtful suggestion to consider a top-down structure similar to concept bottleneck models (CBM). In fact, we have introduced the abstracted top-down model in the introduction and illustrated it in Figure 1(c). However, we believe that structuring the whole paper according to the CBM approach may not effectively showcase our contributions, for the following reasons:
> 1. The structure of our NCG method aligns with the sequence in theoretical causality literature [1] (e.g., “Introduction to Probabilities, Graphs, and Causal Models” -> “Causal Diagrams and the Identification of Causal Effects” -> “Actions, Plans, and Direct Effects”). This structure natually integrates our NCG paradigm into the framework of structural causal models. **This way of writing will make NCG more theoretically logical and make it easier for readers with SCM background to understand.**
> 2.  **We respectfully disagree that our proposed method can be simply regarded as an extension of the concept bottleneck models, as it can not explain some theoritical reasonable designs** including i) Causal effect calculation using PSM and DRL method; ii) Concept reasoning process as equation (6); iii) Intervention training method, which enables effective test-time interventions.
>
> **At this point, we believe that the difficulty of the paper mainly comes from the background knowledge of related techniques. Therefore, in the appendix of the revised manuscript, we will include more detailed explanations and relevant citations of these three components to better describe specific components.**
>
> [1] Causality: Models, Reasoning, and Inference. Judea Pearl
>
> * **R2W2 & R2Q2: Concerning W2, from what perspective do the authors claim that the proposed model is interpretable?**
>
> Thank you for your concern about the interpretability of this work. As you mentioned, we agree that the interpretability of the paper is important, and including the discussion and necessary details can significantly enhance the readability of the paper.
>
> In the context of our paper, the interpretability is demonstrated in several perspective:
> 1. **Causal Graph**. As illustrated in Figure 1(c), we incorporate the causal graph with concepts into the classification process, which introduces interpretability by constructing causal relationships betweeen originally independent posterior concepts and causally related prior concepts.
> 2. **Reasoning Process**. As illustrated in A.5 Case Study (Figure 9), our NCG method can provide **post-hoc interpretability** to see how the reasoning process leads to certain predictions for posterior concepts, which makes the reasoning process more transpanrency. This enhances the transparency of the model, aligning with existing research on GNN interpretability [2].
> 3. **Intervention Process**. Our NCG models not only can perform post-hoc interpretability, but also allows for "intervenable interpreatability" that users can interactively guide the model's reasoning process during testing.
>
> Thanks for your suggestions. We have summarized and added these discussions to the Sections 4.5 in revised manuscript to enhance clarity and understanding:
> > In conclusion, this experiment demonstrates that our NCG method provides post-hoc interpretability, allowing users to understand how the reasoning process leads to specific predictions for posterior concepts. This aligns with existing research on interpretability in graph neural networks [2]. Additionally, our method extends beyond post-hoc analysis by enabling users to actively influence the model’s reasoning during testing through test-time interventions.
>
> [2] Explainability in Graph Neural Networks: A Taxonomic Survey.

---

> ### Author Response · Authors · 2024-11-22
> **Author Rebuttal (II)**
>
> * **R2W3: There were confusing notational errors (e.g., multiple meanings of symbols in the equations).**
>
> Thank you for your careful attention to detail and for highlighting this issue. We have thoroughly reviewed the manuscript and addressed all notational ambiguities to ensure clarity and consistency based on your comments.
>
> * **R2Q3: What was the purpose of the experiments in Section 4.5? If the test samples were intervened and their concepts and labels were perturbed according to a causal graph, it seems natural that the accuracy of the proposed method increased and that of CBM decreased since the former was trained with the causal graph and the latter did not. Can this experiment be considered fair?**
>
> Thanks for your insightful questions. The purpose of the experiments in Section 4.5 is to test whether the NCG classification model can support user-AI interaction through test-time intervention. As discussed in our manuscript (Lines 504 - 512), all test-time intervention experiments were conducted by incrementally intervening on prior concepts, and see how the accuracy of posterior concepts change. **The only difference between NCG and CBM lies in their model structures. Therefore, the comparisons are fair and consistent.** The main purpose of this section is to evaluate whether NCG (with PSM/DRL) and CBM can effectively perform test-time intervention after intervention training—or even without it. We hope this explanation sufficiently clarifies the motivation and fairness of the experiment.

---

> > ### Comment · Reviewer_sJC6 · 2024-11-26
> >
> > I would like to thank the authors for their insightful responses.
> > Because the responses addressed some of my concerns (especially Q2 and Q3), I updated my score.
> >
> > **On R2W1 & R2Q1:**
> > I thank the authors for their clarification. I understand that the proposed method is related to the framework of structural causal models (SCMs). I would appreciate it if the authors would include these discussions when the paper is accepted. (*Additional comment:* I feel it is better to provide a "Preliminaries" section that introduces the basic concept and notation of SCMs before the section on the proposed method. It may help readers unfamiliar with SCMs to understand the proposed framework and its motivation. I hope the authors will consider it if possible. )

---

> > > ### Author Response · Authors · 2024-11-28
> > > **Official Comment by Authors**
> > >
> > > Dear Reviewer sJC6,
> > >
> > > Thank you for your thoughtful feedback and for updating your score based on our responses. We are pleased to hear that our clarifications addressed some of your key concerns, especially regarding Q2 (interpretability) and Q3 (motivations and fairness of experiments in Section 4.5).
> > >
> > > Additionally, we sincerely appreciate your constructive suggestion to include a "Preliminaries" section introducing the basics of structural causal models. We have carefully considered this feedback and included a dedicated "Preliminaries" section in the revised manuscript (highlighted in blue).
> > >
> > > We believe this addition significantly enhances the paper’s clarity and accessibility for readers unfamiliar with SCMs, and we deeply appreciate your guidance in this regard. If you have any additional questions or concerns, we would be more than happy to address them. Your feedback has been instrumental in improving the clarity and depth of our paper. Thank you again for your time, thoughtful comments, and engagement with our work.
> > >
> > > Best regards,
> > >
> > > Authors

---

### Official Review · Reviewer_bMLv · 2024-11-03

**Soundness:** 3
**Presentation:** 4
**Contribution:** 3
**Rating:** 8
**Confidence:** 3

**Summary:**

This paper combines the causal inference frameworks of structural causal models and potential outcomes models, using neural network implementations, to provide better transparency and interactive reasoning capabilities in classification settings. Estimation of causal effects (ATEs) to build the underlying graph is is done using propensity score matching and doubly robust learning. Compared to naive utilization of neural networks for classification, this approach gives a more inherently interpretable model for classification by explicitly modeling underlying causal relationships. The authors demonstrate their method on the ImageNet and Bird datasets and show improved performance as well as improved interpretability and human intervene-ability (via test-time intervention experiment) over certain neural network baseline models. They argue it is the explicit modeling and tweaking of underlying causal structure that gives accuracy improvements, with some ablations as evidence.

**Strengths:**

Very clear intro and stated contributions. Work is original and well-motivated. Results feel complete and experiments are convincing (e.g. with ablations showing the importance of both Intervention Training and Learnable Scaled Weight). As a reader, most of my immediate/obvious questions are answered in the paper and/or the Appendix.

Strong conceptual figures; I am familiar with traditional causal inference frameworks and appreciated the concise and relevant presentation here. Tables and graphs are appropriately used; the one on ablations is especially clear. The Appendix provides clear pseudocode and more concrete examples and visualizations of the authors’ method and experiments – relevant info to better understand the model and application concretely.

Represents a solid and clear contribution and advance for a conference paper work. Above acceptance threshold.

**Weaknesses:**

I have no major issues with this paper.

Small typo: 310/311 intervention is spelled wrong.

More of a leading question: is the prior work complete? It's a relatively small section of the Intro, and I wonder if there are previous attempts to integrate causal inference frameworks in NNs (maybe not in interpretability specifically) that are important to acknowledge and relevant to cite. I.e. previous work on modeling causality with GNNs? Your paper does not obviously acknowledge that causal reasoning on graphs implemented as NNs is an established concept. You say "Unlike conventional methods that rely on correlation-based learning, NCG constructs a causal graph..." but GNNs attempting to model causality aren't mentioned.

**Questions:**

Is there anything systematic that you can comment on or hypothesize about in Section 4.2 about DRL doing better on Bird vs. PSM doing better on ImageNet performance in Table 2?

You are space constrained, but since the intervention training method is an important contribution, I wonder if it could get a little more space in the main text.

Regarding your explanation of the accuracy drop for DRL+CLIP w/o intervention training: are you saying that the method reaches a point (high number of nodes) after which it can’t learn the underlying structure (w/o intervention training) and therefore reason appropriately (“get the right answer”)? Are there metrics other than accuracy that could demonstrate the difference in underlying reasoning (even just at the point on the graph, ~1300 nodes for CLIP+DRL, between the two graphs) and make your claim stronger?

---

> ### Author Response · Authors · 2024-11-22
> **Author Rebuttal (I)**
>
> # To Reviewer bMLv (R1)
>
> * **R1W1: I have no major issues with this paper.**
>
> We sincerely thank you for your positive feedback and appreciation of the strengths of our work. We are pleased to know that you found the introduction clear, the contributions well-articulated, and the experiments convincing. Your recognition of the importance of our ablation studies in demonstrating key aspects of our methodology is particularly encouraging, and we greatly appreciate your specific suggestions regarding our intervention training method to enhance the important contributions of the manuscript. We are committed to ensuring that the final manuscript maintains this clarity and rigor. Thank you again for your supportive and constructive review!
>
> * **R1W2: Small typo: 310/311 intervention is spelled wrong.**
>
> We appreciate your attention to detail! We have revised the spelling of "intervention" on Lines 310/311 according to your feedback.
>
> * **R1W3: Suggested clarification on the causal inference background, specifically to include prior work integrating causal inference in neural networks, especially in GNNs.**
>
> Thank you for this valuable suggestion. We acknowledge that our initial description of causality-driven GNNs could be improved. To address this, we have revised the manuscript to clarify our contributions and to better highlight how our work differs from previous studies.
>
> 1. Our research mainly aims to enhance interpretable and intervenable classification capability of neural networks. However, most of causality-driven GNNs are not designed for this purpose.
> 2. There are three types of previous research regarding the causal learning on graphs: 1) Causal reasoning; 2) Causal discovery; 3) Causal representation learning. These tasks mainly focus on interpretability of data by quantifying cause-effect relationships among variables or recovering the causal graph from an observed dataset, rather than using the reasoning ability of an structural causal model for intervenable classification.
> 3. To address your comments, we have revised the manuscript and added clarification in the Introduction as follows:
> > Furthermore, causality-driven GNNs [1, 2] focus on interpreting data by quantifying cause-effect relationships among variables or recovering the causal graph from observed datasets. However, these models lack the capability for interactive intervention and do not support interactive classification involving human users.
> > To address these challenges, we propose the Neural Causal Graph (NCG), a novel framework that enable intervenable classification by integrating causal inference within neural networks architectures. Unlike conventional methods that can only perform post-hoc interpretable predictions, NCG constructs a causal graph that models the underlying mechanisms of the data, enabling the model not only can reason by following the causal relationships between labels, but also supports intervenable classification within reasoning process.
>
> We hope the clarifications provided will adequately address your concern.
>
> [1] When Graph Neural Network Meets Causality: Opportunities, Methodologies and An Outlook
>
> [2] Exploring Causal Learning through Graph Neural Networks: An In-depth Review
>
> *	**R1Q1: Is there anything systematic that you can comment on or hypothesize about in Section 4.2 about DRL doing better on Bird vs. PSM doing better on ImageNet performance in Table 2?**
>
> Thanks for bringing up this specific experimental phenomenon to our attention. The observed performance difference likely stems from differences in weight estimation strategies used by the PSM and DRL methods. DRL performs better on the smaller Bird dataset, while PSM shows superior results on the larger ImageNet dataset. This observation aligns with previous findings [3] that PSM tends to estimate causal effects more accurately when the treatment and control groups are larger. In fact, since some of the performance difference between PSM and DRL on each dataset are relatively small, it is hard to concrete hypothesize based on this uncertain phenomenon.
>
> [3] Matching methods for causal inference: A review and a look forward.

---

> ### Author Response · Authors · 2024-11-22
> **Author Rebuttal (II)**
>
> *	**R1Q2: You are space constrained, but since the intervention training method is an important contribution, I wonder if it could get a little more space in the main text.**
>
> Thanks for your recognition of the importance of our proposed intervention training method, we have expanded the intervention training mechanism in revised paper, outlining the methodology more explicitly.
> > The intervention training method is designed to enhance the model’s reasoning ability by aligning the training and testing distributions, ensuring robustness under test-time interventions. Inspired by the principles of structural causal models (SCM), this approach simulates the effects of the $ do(\cdot) $ operator during training, isolating the influence of intervened nodes by eliminating their dependence on parent nodes. Specifically, during each training iteration, a subset of prior concepts is randomly selected with an intervention rate $ p = 0.15 $, and their values are fixed using the formula:
> $$
> Z_i = (2Y_i - 1) \times v,
> $$
> where $ Z_i $ is the intervened node value, $ Y_i \in \{0, 1\} $ represents the corresponding multi-label value, and $ v $ is an empirically determined confidence value set to $ 5 $. The sigmoid output of $ -5 $ or $ 5 $ reflects high certainty about the node’s intervention status. This process is designed to enable the model to address "What if" questions by learning the dynamics among concepts through intervention. By simulating test-time scenarios, the model is conditioned to predict posterior concepts based on both inferred and intervened prior concepts. An overview of the procedure for intervention training is detailed in Algorithm 2 (Appendix).

---

> ### Author Response · Authors · 2024-11-22
> **Author Rebuttal (III)**
>
> * **R1Q3: Regarding your explanation of the accuracy drop for DRL+CLIP w/o intervention training: are you saying that the method reaches a point (high number of nodes) after which it can’t learn the underlying structure (w/o intervention training) and therefore reason appropriately (“get the right answer”)? Are there metrics other than accuracy that could demonstrate the difference in underlying reasoning (even just at the point on the graph, ~1300 nodes for CLIP+DRL, between the two graphs) and make your claim stronger?**
>
> Thank you for this valuable observation. The accuracy drop after around 1300 nodes likely arises because, without intervention training, the model can not **reliably** learn causal dependencies of graph, particularly as the number of nodes grows. It took us a bit longer than expected to carefully examine the experiments with DRL+CLIP by gathering all the necessary details, and there are three main reasons that, in an unexpected combination, led to this phenomenon when the model lacks interventional training. Note that our proposed NCG models with intervention training do not suffer from this issue, showing the necessity of our proposed intervention training process.
>
> 1. **Positive Inductive Bias in the Causal Graph**: The model inherently interprets positive logits in the formula $ C_j = \phi(g(h(X))\_{j} + \sum w_{ij} \cdot C\_i + U\_j) $ as an indication of the presence of certain concepts in the images. This is a reasonable assumption given the structure of our causal graph, which has an inductive bias toward positive causal relationships. As detailed in Section 3.2.1 "Causal Effect Calculation", the causal effect weights are typically estimated with 1 representing the presence of a concept and 0 representing its absence. This means that the estimated causal edges generally have positive weights, leading to an increase in the logits of child nodes when the logits of the corresponding parent concepts increase.
>
> 2. **Negative Causal Weights in DRL Estimation**: Although most of the estimated edge weights are positive, there are few exceptions where the causal weights are negative. This occurs because the DRL method uses a regression-based estimator (Lines 240-258, Equation (4) and (5)) to calculate causal effects, and in some cases, this results in a negative weight. When these negative weights exist, to increase the logits of their child nodes, the model must reduce the logits of parent concepts.
>
> 3. **Intervention on the Last 100 Prior Concepts Close to Posterior Concepts**: We observed that the last 100 prior concepts in the causal graph are most closely related to the posterior concepts within one-hop. During the intervention, we used values of -5 and 5 to manipulate the prior concepts (Section 4.5 Test-time Intervention). In extreme cases where the causal weights are negative, applying a -5 intervention to the prior concepts was mistakenly interpreted by the model as an increase in the corresponding posterior concept logits. This misinterpretation possibly leads to wrong classification results, thus causing a sudden and significant drop in accuracy.
>
> **These factors combined to produce the drastic drop in model performance after the initial 1300 effective node interventions. Note that this situation arises because the model did not perform intervention training, and the fact that the model was able to improve performance on the first 1300 nodes is already a noteworthy result. Conversely, models that have undergone proper intervention training do not suffer from this issue, demonstrating the effectiveness of our approach. Thank you very much for pointing out this problem. Your feedback provides valuable insights that led to a more thorough investigation. We will include these detailed explanations in the appendix of the revised paper.**

---

> > ### Comment · Reviewer_bMLv · 2024-11-26
> > **Thanks for the responses and additional details**
> >
> > Thank you for addressing my comments and questions in detail! I maintain my score of 8.

---

> > > ### Author Response · Authors · 2024-11-28
> > > **Official Comment by Authors**
> > >
> > > Dear Reviewer bMLv,
> > >
> > > Thank you for your kind feedback and for confirming that our response addressed your concerns. We sincerely appreciate your positive comments and valuable suggestions. Please do not hesitate to share any additional ideas or insights as we continue engaging with the other reviewers. Once again, thank you for your exceptional support and for recognizing the value of our work!
> > >
> > > Best regards,
> > >
> > > Authors

---

### Author Response · Authors · 2024-11-22
**To All Reviewers**

# To All Reviewers

We are grateful to all four reviewers for your constructive comments and appreciation of our work’s strengths. Specifically, we thank you for acknowledging the novelty of combining causal inference with neural networks **bMLv** (**R1**), **sJC6** (**R2**), **ytQg** (**R3**), **CUVg** (**R4**), the contributions to the field of intervenable AI **R1**, **R3**, and the promising empirical results shown in our experiments **R1**, **R3**, **R4**. We appreciate your recognition of the clarity of our writing **R1**, and the potential for practical HCI context **R3**.

Your feedback is invaluable for refining and improving our work. We have addressed the major suggestions and revised our manuscript. We believe these revisions will enhance the manuscript’s contribution to the field of interpretable and intervenable classification. In this response, we address each comment individually and clarify any points of confusion. Each question is labeled RiQj or RiWj, where i is the reviewer number and j is the question number or weakness number.

---

### Meta-Review · Area_Chair_MKk3 · 2024-12-11

**Metareview:**

In this work, the authors propose a combination of causal inference and neural networks to provide interpretable and intervenable classification. Their framework is evaluated on 2 datasets, showing a small but significant improvement over baselines. Ablation studies are performed, and the authors have added a pilot study on interpretability with 5 human users.

This paper feels borderline to me due to the limited improvement in performance and limited quantitative evaluation of the potential benefits of such an architecture. However, I believe this is an interesting contribution to the research avenue combining causal inference and neural networks and will hence propose acceptance.

**Additional Comments On Reviewer Discussion:**

The authors engaged in the discussion and provided a revised version of the paper. They pushed back on some comments regarding the organization of the paper. Having a look myself, I did not identify a need for changing the paper structure, but still encourage the authors to amend their text to provide a high-level intuition before diving into the details of the method.

---

### Decision · Program_Chairs · 2025-01-22

Accept (Poster)